# WebSailor-V2: Bridging the Chasm to Proprietary Agents via Synthetic Data and Scalable Reinforcement Learning

Kuan Li[1,3][*][†], Zhongwang Zhang[2,3][*], Huifeng Yin[3][*][†], Rui Ye[3][*], Yida Zhao[3][*], Liwen Zhang[3][*], Litu Ou[3], Dingchu Zhang[3], Xixi Wu[3], Xinmiao Yu[3], Jialong Wu[3], Xinyu Wang[3], Zile Qiao[3], Zhen Zhang[3], Yong Jiang[3][†], Pengjun Xie[3], Fei Huang[3], Zhi-Qin John Xu[2][†], Shuai Wang[1][†], Minhao Cheng[4][†], Jingren Zhou[3]

[1] The Hong Kong University of Science and Technology
[2] School of Mathematical Sciences, Institute of Natural Sciences, Shanghai Jiao Tong University
[3] Tongyi Lab, Alibaba Group
[4] Penn State University

## Abstract

To significantly advance the capabilities of open-source web agents, we present WebSailor-V2, a complete post-training pipeline encompassing data construction, Supervised Fine-Tuning (SFT), and Reinforcement Learning (RL). Our methodology features two key innovations: (1) On the data front, we developed SailorFog-QA-2, a novel dataset built from a densely interconnected knowledge graph that introduces a wide variety of uncertainties beyond simple obfuscation, fostering more sophisticated reasoning. (2) For training, we engineered a dual-environment RL framework, combining a high-fidelity simulator for rapid, low-cost algorithmic iteration with a robust, managed real-world environment for stable final policy training, all integrated within a symbiotic data-policy feedback loop. Trained on the Qwen3-30B-A3B model, WebSailor-V2 achieves state-of-the-art results, scoring 35.3 on BrowseComp-EN, 44.1 on BrowseComp-ZH, and 30.6 on Humanity's Last Exam (HLE). Notably, our 30B-A3B MOE agent significantly outperforms all existing open-source agents and surpasses even the 671B DeepSeek-V3.1, demonstrating performance competitive with leading proprietary systems.

## 1 Introduction

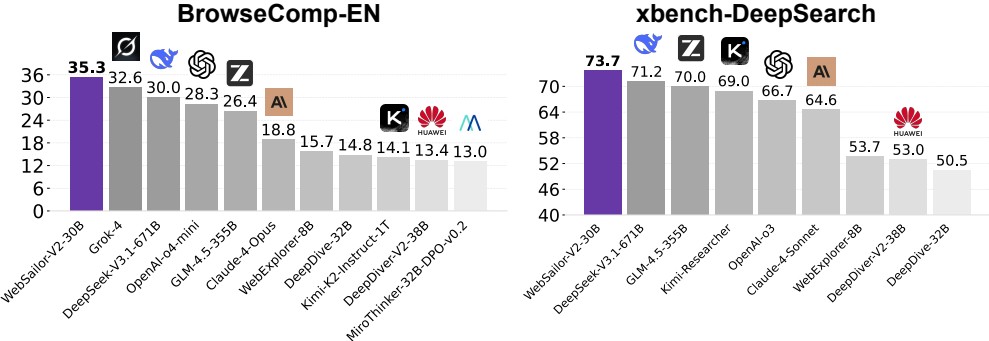

Figure 1: Performance on the BrowseComp-EN and xbench-DeepSearch benchmarks.

In the pursuit of Artificial General Intelligence (AGI), autonomous AI agents represent a critical milestone, with "Deep Research" emerging as a core paradigm for achieving more generalized

---

[*]Equal contribution (the first six authors contributed equally).

[†]Corresponding authors: kuanli@unipat.ai, yinhuifeng.andy@gmail.com, yongjiang.jy@alibaba-inc.com, xuzhiqin@sjtu.edu.cn, shuaiw@cse.ust.hk, mmc7149@psu.edu

capabilities. By leveraging external tools like search engines and web browsers, these agents can autonomously conduct systematic and in-depth analyses to tackle complex, multi-step research tasks through dynamic reasoning and iterative information retrieval (OpenAI, 2025a; AI, 2025). Despite recent advancements across the research community, spanning improvements from both perspectives of data and training (Wu et al., 2025b; Li et al., 2025b; Liu et al., 2025a; Nguyen et al., 2025; Li et al., 2025c; Wu et al., 2025a; Tao et al., 2025), a considerable performance gap still persists between open-source solutions and proprietary systems (e.g., OpenAI DeepResearch (OpenAI, 2025a)), leading to a bottleneck in democratizing powerful research capabilities.

This performance disparity primarily stems from fundamental challenges in two of the most critical stages for developing powerful agents: data and training. (1) **Data: insufficient diversity and monolithic definitions of uncertainty.** Information-seeking relies on the agent's ability to leverage existing information and logical relationships to infer or acquire new, reliable knowledge. If the training data lacks a sufficiently broad and complex range of logical structures, the model will struggle to generalize to novel and intricate problems. Existing methodologies often rely on a narrow set of uncertainty definitions, such as obfuscation (Li et al., 2025b; Gao et al., 2025; Shi et al., 2025). A wider variety of uncertainty types is needed to elicit more diverse and sophisticated reasoning behaviors from the base model, better preparing it for the ambiguity inherent in real-world research. (2) **Training: lack of scalable reinforcement learning (RL) training environment.** Creating a scalable and robust RL training environment for agentic systems poses a significant challenge, which typically demands massive rollouts, each potentially involving numerous tool calls. The high cost and engineering complexity of high-concurrency requests to external APIs can lead to practical issues like tool latency, API failures, and inconsistent outputs. These issues would contaminate the training data, degrade the model's learned policies, and severely hinder iteration of RL training algorithms.

In this paper, we present WebSailor-V2, a comprehensive framework designed to overcome these barriers through topological data synthesis and environmental stabilization. **(1) Topological Strategy for Reasoning:** To resolve the insufficient diversity inherent in linear or tree-like training data, we introduce **SailorFog-QA-V2**, an enhanced dataset built upon SailorFog-QA (Li et al., 2025b). Unlike conventional "easy-to-hard" expansion, our method constructs densely interconnected knowledge graphs. By specifically sampling trajectories that require resolving cyclic dependencies and traversing critical "cut vertices", we force the agent to learn abstract graph-search patterns. This mechanism ensures the model acquires robust reasoning capabilities that generalize beyond simple information retrieval. **(2) Stabilization Strategy for RL:** To mitigate the "signal-to-noise" ratio issue caused by real-world volatility, we propose a **Symbiotic Dual-Environment framework**. By utilizing a high-fidelity simulator as an algorithmic "Wind Tunnel", we decouple policy learning from environmental stochasticity. This allows for high-frequency, stable policy iteration, ensuring that the agent establishes a robust policy before adapting to the noisy, managed real-world interface. First, we develop a dedicated **simulated environment** from the ground up, based on a large-scale offline Wikipedia knowledge base (Vrandečić & Krötzsch, 2014). This environment is designed for high-frequency algorithmic experimentation and data curation, providing a low-cost, exceptionally fast, and fully controllable platform. Through meticulous design, it achieves high fidelity, ensuring that the agent's interaction dynamics, state transitions, and reward mechanisms closely mirror those of a real-world setting. Second, recognizing that RL training in a real environment is a complex engineering problem—especially concerning the consistency of tool returns after toolset expansion, the reproducibility of trajectory sampling, and the need for high concurrency and fault tolerance, we build a unified tool execution interface that utilizes a scheduling and management layer to incorporate different tools' quality measures and protocols. Finally, our data construction and RL training pipelines are integrated into a symbiotic feedback loop. This dynamic mechanism allows the system to synthesize and filter high-quality data based on training dynamics, enabling the model to continually refine its policies and learn from a stream of relevant information. This co-evolution of data and policy therefore promotes building deep research agents more effectively and efficiently.

To demonstrate the efficacy of SailorFog-QA-V2 and training strategies, we build our agent upon the foundational ReAct framework (Yao et al., 2023). Trained on Qwen3-30B-A3B (Yang et al., 2025), our WebSailor-V2-30B-A3B achieves scores of 35.3 on BrowseComp-EN (Wei et al., 2025) and 44.1 on BrowseComp-ZH (Zhou et al., 2025a), alongside a score of 30.6 on HLE (Phan et al., 2025), significantly outperforming all existing agents built on open-source models. Remarkably, our 30B-sized agent outperforms the previous best-performing agentic 671B-sized LLM DeepSeek-V3.1 (Team, 2025b), which achieves 30.0 on BrowseComp-EN and 29.8 on HLE, respectively.

## 2 AGENTIC FRAMEWORK

**ReAct.** We adopt the ReAct framework as the foundation for our agent's architecture (Yao et al., 2023):

$$\mathcal{H}_T = (\tau_0, a_0, o_0, \dots, \tau_i, a_i, o_i, \dots, \tau_T, a_T), \tag{1}$$

where $\tau_i$, $a_i$, $o_i$ represent **thought**, **action**, and **observation** in the $i$-th iteration, respectively. At step $t$, the agent's thought $\tau_t$ and subsequent action $a_t$ are sampled from a policy $\pi$ conditioned on the complete preceding context, defined as $\pi(a_t, \tau_t | \mathcal{H}_{t-1})$.

While more complex single and multi-agent paradigms have emerged, our choice of ReAct is a deliberate one, rooted in its simplicity and alignment with fundamental principles. This decision is heavily informed by "The Bitter Lesson" (Sutton, 2019), which posits that general methods leveraging scalable computation ultimately outperform approaches that rely on complex, human-engineered knowledge and intricate designs. Frameworks that require extensive, specialized prompt engineering or possess rigid operational structures risk becoming obsolete as the intrinsic capabilities of models scale (Li et al., 2025a).

**Action space.** The action space is composed of four primary tools: search, visit, Google Scholar, and Python interpreter, along with the terminal action final answer. Details of tools are provided in the Appendix C.

## 3 SAILORFOG-QA-V2

This section focuses on the data construction of SailorFog-QA-v2, where we introduce how we construct a dense knowledge graph containing real internet information and how we generate question-answer (QA) pairs based on this data structure.

### 3.1 GRAPH CONSTRUCTION

An information retrieval problem, at its core, can be conceptualized as navigating a complex web of entities and their interrelationships. To effectively address such problems, especially in the context of advanced AI agents performing "Deep Research", it is crucial for models to comprehend and leverage these underlying structural connections. Therefore, to ensure our generated QA pairs encompass a rich and diverse spectrum of logical relationships, our foundational approach involves constructing a comprehensive knowledge graph. This graph serves as a robust substrate from which we can sample various structurally distinct subgraphs, each forming the basis for generating questions that probe different reasoning patterns.

Recent advancements in data construction for web agents have also aimed at acquiring such structured information. These methods typically initiate from a simple "seed" question, progressively expanding the graph by employing external tools (e.g., search or browsing) to discover related entities and facts (Gao et al., 2025; Liu et al., 2025a; Tao et al., 2025). However, a significant drawback of this "easy-to-hard" or iterative expansion strategy is its inherent tendency to produce predominantly tree-like or acyclic logical structures. While effective for certain types of information retrieval, this approach inherently struggles to capture or generate scenarios involving complex cyclic relationships, feedback loops, or intricate interdependencies that are common in real-world knowledge graphs.

Building upon the foundational framework of SailorFog-QA (Li et al., 2025b), V2 still starts with a seed entity and leverages web tools to discover related entities and extract their corresponding information. However, to achieve a more comprehensive topological coverage to overcome the limitations of acyclic graphs,we introduce significant enhancements to the graph expansion phase. Specifically, we actively seek out and establish more dense connections between nodes, intentionally creating cyclic structures. This ensures that the resulting graph is not merely a sprawling tree but a richly interconnected web, more accurately reflecting the complex, non-linear nature of real-world knowledge. Beyond these structural improvements, we now preserve more complete procedural information, such as the specific search queries used and the source URLs that led to a new discovery. Furthermore, we compute and store various statistical features for each entity, which are instrumental for the subsequent QA generation phase, enabling us to craft more nuanced and challenging questions.

## 3.2 SUBGRAPH EXTRACTION

In the previous version, our subgraph sampling strategy relied on random sampling, with an attempt to enumerate all possible substructures of a fixed edge count. However, as the graph in V2 has become substantially denser, such an exhaustive enumeration is computationally infeasible due to combinatorial explosion. To overcome this scalability issue, we adopt a random-walk based approach for subgraph extraction. Ultimately, this strategy enables us to efficiently gather a sufficient quantity of non-isomorphic (verified by Weisfeiler-Leman algorithm (Weisfeiler & Leman, 1968)), connected subgraphs that collectively represent the full spectrum of structural complexities, without the prohibitive cost of a brute-force search.

## 3.3 QA GENERATION

When generating QA, we do not directly feed the subgraph into the LLM end-to-end to produce the result. Instead, we first analyze how many non-isomorphic nodes exist in a given topology, so that the QA focus can be evenly distributed across all orbit nodes (i.e., nodes that occupy different structural roles). Moreover, obfuscation has become one of the most common methods for introducing uncertainty and eliciting high-order reasoning patterns in the construction of challenging information-seeking tasks (Li et al., 2025b; Gao et al., 2025; Shi et al., 2025; Liu et al., 2025a; Geng et al., 2025). Specifically, obfuscation corresponds to the reasoning behavior required when a query's key elements—such as specific entities, dates, or values—are replaced with more general or ambiguous descriptions. Answering such questions compels the model to move beyond simple keyword matching, engaging in contextual inference to disambiguate underspecified entities, generating and verifying hypotheses through iterative information gathering, and synthesizing evidence from multiple sources to converge on a conclusive answer. However, this set of skills, while crucial, represents only a subset of the capabilities required for a truly super-human web agent. To this end, we introduce a wider array of defined uncertainties beyond simple obfuscation. These include: (1) Semantic Ambiguity: We deliberately under-specify key entities (e.g., "the mathematician who generalized this result" instead of the specific name) or dates. This forces the agent to reason over the graph structure to disambiguate entities via context rather than keyword matching. (2) Distractor Noise: We inject plausible but factually incorrect distractors into the context (e.g., a publication year of a related paper by the same author). This requires the agent to perform active cross-verification and contradiction detection. (3) Structural Constraints: By analyzing non-isomorphic nodes—nodes occupying distinct structural roles—we identify critical paths (e.g., bridges or cut vertices). We generate questions that require traversing these specific "cut" edges, ensuring the agent engages in global graph exploration rather than local greedy search.

## 3.4 DATASET STATISTICS

The resulting SailorFog-QA-V2 dataset used for SFT and RL consists of over 30,000 high-quality instruction-tuning pairs. The underlying knowledge graphs are densely connected, with an average of approximately 30 nodes and an average degree of 2.5 per connected component. The random-walk sampling strategy, verified by the Weisfeiler-Leman algorithm, ensures a diverse coverage of topological structures, ranging from simple chains to complex cycles and dense clusters.

## 4 AGENTIC POST-TRAINING

## 4.1 SFT COLD START

The initial phase of our agentic post-training pipeline is a Supervised Fine-Tuning (SFT) stage, designed to provide the base model with a robust initial policy before the commencement of reinforcement learning. To ensure a controlled and high-quality training regimen, our SFT dataset is constructed entirely from synthetic data derived from the SailorFog-QA-V2 generator. The training trajectories are produced by high-performing, open-source models solving the generated QA tasks, with quality maintained via rejection sampling. In a key architectural decision, and a departure from prior work like WebSailor, our agent is built upon the Qwen3-30B-A3B-Thinking-2507 foundation model (Yang et al., 2025), with the context length deliberately extended to 128k.

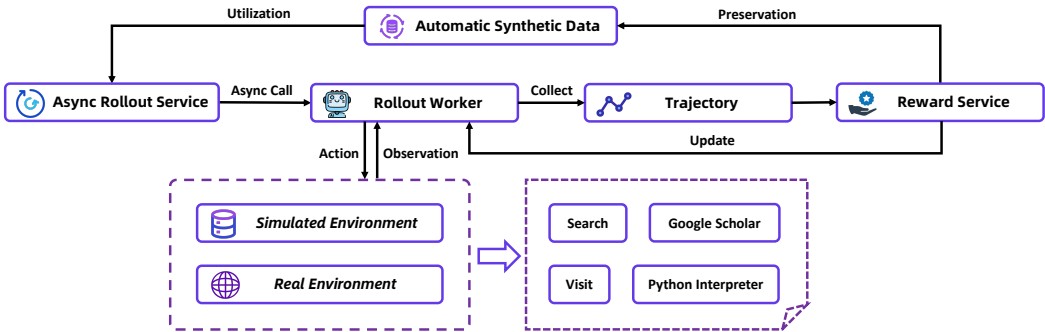

Figure 2: An overview of our Reinforcement Learning framework. The agent is trained in a closed loop where the policy is continuously updated through interactions with simulated or real-world environments. A key component is the automated data synthesis and filtering pipeline, which dynamically curates training data based on the training dynamics.

## 4.2 AGENTIC REINFORCEMENT LEARNING

Our RL strategy employs a dual-environment approach, leveraging the benefits of both controlled simulation and real-world deployment.

**Simulated Environment.** The practice of training in simulation for subsequent policy transfer or algorithm validation is a common and essential strategy in research and development (Da et al., 2025), successfully applied across domains like robotics and perception (Osiński et al., 2020; Haiderbhai et al., 2024; Ho et al., 2021). Reliance on commercial real-world web APIs (e.g., SerpAPI (SerpAPI, 2025) or Jina (Jina.ai, 2025)) introduces significant practical constraints, including high operational costs, restricted Queries Per Second (QPS), and output inconsistencies. During the critical initial stages of algorithm research and data curation, these real-world limitations can severely decelerate the development cycle and compromise the reliability of ablation studies.

To circumvent these issues, we constructed a fully controllable simulated environment utilizing an offline Wikipedia database and a corresponding suite of simulated web tools. We adapted the SailorFog-QA-V2 generation pipeline to operate exclusively on this offline corpus, thereby synthesizing a dedicated, structurally complex training and testing dataset that is perfectly aligned with the simulation's capabilities. This methodology provides a cost-efficient and fully observable platform, significantly accelerating our algorithmic experimentation and iteration process.

We utilize this simulator not as a direct content replica of the live web, but as a "wind tunnel" for algorithmic stability. Our internal validation demonstrates a high correlation in training dynamics (specifically Reward and Pass@1 trends) between the simulator and the real environment. Algorithmic improvements that stabilize learning in the simulator consistently translate to stability in the real world. Consequently, we adopt a staged training strategy: we perform high-frequency hyperparameter tuning and reward shaping validation in the simulator, and then execute the final production training run in the managed real-world environment using the validated configuration.

**Real Environment.** While simulation is invaluable for rapid prototyping, the ultimate objective is real-world performance. Transitioning to a real environment, however, introduces considerable engineering challenges. Our agent relies on a multifaceted toolkit integrating various search sources, diverse webpage parsers, and a code execution sandbox. The inherent volatility of external APIs (e.g., latency, failure, or inconsistent returns) within this composite system poses a significant risk of data contamination, which can obscure the true source of performance degradation—making it difficult to isolate algorithmic deficiencies from environmental instability. To ensure stability, we architected a robust, unified tool execution interface. This interface includes a scheduling and management layer that orchestrates tool execution and incorporates sophisticated concurrency handling and fault-tolerance mechanisms. These include QPS constraints, result caching, automated timeout-and-retry protocols, service degradation strategies for non-critical failures, and seamless switching to backup data sources. This multi-layered design effectively abstracts the tool invocation process into a

deterministic and stable interface from the agent's perspective, successfully insulating the training loop from real-world stochasticity while simultaneously optimizing operational costs.

**Data Curation.** Data quality is the central driver of enhanced model capability; its importance often surpasses that of the algorithm itself. The quality of the training data directly establishes the upper bound for the model's ability to generalize to out-of-distribution scenarios through self-exploration. To address this, we implemented a data optimization loop guided by real-time training dynamics. This optimization is achieved via a fully automated data synthesis and filtering pipeline that dynamically curates the training set. By establishing this closed loop between data generation and model training, our approach not only ensures training stability but also yields substantial performance gains by continually feeding the model with the most informative trajectories.

**RL algorithm.** Our RL algorithm is a tailored adaptation of GRPO (Shao et al., 2024):

$$
\begin{aligned}
\mathcal{J}(\theta) = \quad & \mathbb{E}_{(q,y)\sim\mathcal{D},\{o_i\}_{i=1}^G\sim\pi_{\theta_{\text{old}}}(\cdot|context)} \\
& \left[ \frac{3}{\sum_{i=1}^G |o_i|} \sum_{i=1}^G \sum_{t=1}^{|o_i|} \min\left( r_{i,t}(\theta)\hat{A}_{i,t},\ \text{clip}\left( r_{i,t}(\theta), 1-\varepsilon_{low}, 1+\varepsilon_{high} \right)\hat{A}_{i,t} \right) \right],
\end{aligned}
\tag{2}
$$

where $(q, y)$ is the question-answer pair, $r_{i,t}(\theta)$ is the importance sampling ratio, and $\hat{A}_{i,t}$ is an estimator of the advantage at time step $t$:

$$
r_{i,t}(\theta) = \frac{\pi_\theta(o_{i,t} \mid context)}{\pi_{\theta_{\text{old}}}(o_{i,t} \mid context)}, \quad \hat{A}_{i,t} = R_i - \text{mean}(\{R_i\}_{i=1}^G).
\tag{3}
$$

We employ a strictly on-policy training regimen, where trajectories are continuously sampled using the most up-to-date policy, ensuring that the learning signal is always relevant to the model's current capabilities. Following DeepSwe (Luo et al., 2025) and DAPO (Yu et al., 2025b), the training objective is optimized using a token-level policy gradient loss. Second, to further reduce variance in the advantage estimation, we adopt a leave-one-out strategy (Chen et al., 2025). Furthermore, we employ a conservative strategy for negative samples, having observed that an unfiltered set of negative trajectories significantly degrades training stability. This can manifest as a "format collapse" phenomenon after extended training. To mitigate this, we employ a targeted filtering strategy. We selectively mask the loss for trajectories that exceed the length limit without yielding a final answer, as these provide ambiguous learning signals. In contrast, trajectories containing explicit format errors or invalid tool calls are assigned negative rewards rather than being masked, allowing the model to actively learn to avoid such behaviors. This strategy effectively prevents format collapse while maintaining training stability. For the sake of efficiency, we do not employ dynamic sampling. We instead leverage larger batch and group sizes, which serve to maintain smaller variance and provide adequate supervision.

**However, we consider that the algorithm is important but not the only decisive factor in the success of Agentic RL. We have experimented with many different algorithms and tricks, and find that data and stability of the training environment are likely the more critical components in determining whether the RL works. Interestingly, we have tested to train the model directly on the BrowseComp testing set, but the results are substantially poorer than when using our synthetic data. We hypothesize that this disparity arises because the synthetic data offers a more consistent distribution, which allows the model to be more effectively tailored. Conversely, the human-annotated data (such as BrowseComp) is inherently noisier. Given its limited scale, it is difficult to approximate a learnable underlying distribution, which consequently hinders the model to learn and generalize from it.**

## 5 EXPERIMENTS

### 5.1 SETUP

**Research Questions** To verify our methodological claims, our evaluation is designed to answer two core scientific questions:

- **(RQ1) Topological Generalization:** Does training on data derived from dense, cyclic knowledge graphs (SailorFog-QA-V2) yield stronger reasoning capabilities compared to standard linear or tree-like data expansion used by baselines?
- **(RQ2) Environmental Stability:** Does decoupling training into a dual-environment (Simulation-to-Real) framework effectively mitigate the instability of real-world RL and lead to better convergence?

**Models and Benchmarks** We perform SFT and RL training on Qwen3-30B-A3B-2507 (Yang et al., 2025). We mainly evaluate our method on six representative and challenging benchmarks:

- **BrowseComp-EN** (Wei et al., 2025): One of the most challenging benchmarks introduced by OpenAI to evaluate the proficiency of AI agents in locating hard-to-find, often multi-faceted, information across the internet, which demands sophisticated browsing strategies and reasoning capabilities.
- **BrowseComp-ZH** (Zhou et al., 2025a): Similar to BrowseComp-EN, but the QAs are in Chinese.
- **GAIA** (Mialon et al., 2023): A benchmark that requires multi-modality and tool-use abilities. We only use a subset of 103 cases from the text-only validation subset (Li et al., 2025c; Wu et al., 2025a).
- **xbench-DeepSearch** (Xbench-Team, 2025): A new, dynamic, professionally-aligned benchmark that focuses on evaluating AI agents' tool usage capabilities, specifically in deep information retrieval and complex search tasks.
- **Humanity's Last Exam (HLE)** (Phan et al., 2025): HLE is a global collaborative effort, with questions from nearly 1,000 subject expert contributors affiliated with over 500 institutions across 50 countries – comprised mostly of professors, researchers, and graduate degree holders.
- **DeepResearch Bench** (Du et al., 2025): This benchmark is comprised of numerous PhD-level research tasks designed to evaluate the performance of deep-research agents, specifically focusing on the quality of their generated research reports and their proficiency in information retrieval and collection.

**Baselines** For proprietary models accessed via API or manual testing, we maintained consistent evaluation conditions to ensure fairness, enforcing a 128k context token limit and a maximum budget of 100 tool calls, consistent with our agent's settings. We compare our method with the following paradigms:

- **Proprietary Browsing Agents**: We test Gemini-2.5-pro-DeepResearch (Team, 2025c), Claude-Research (Team, 2025a), Doubao-Deepresearch (Doubao, 2025), Perplexity-Research (Team, 2025g), Grok-Deeper-Search (Team, 2025d), Claude-4-Sonnet (anthropic, 2025), OpenAI-o3 (OpenAI, 2025b), OpenAI DeepResearch (OpenAI, 2025a); however, as not all of them are fully accessible via API, they were not tested across all benchmarks and experiments.
- **Open-Source Agents**: We compare our method with recent open-source web/search agents, including ASearcher-Web-QwQ (Gao et al., 2025), MiroThinker-32B-DPO-v0.2 (Team, 2025e), WebSailor-72B, WebExplorer-8B, DeepDiver-V2-38B (Team, 2025f), DeepDive-32B (Lu et al., 2025), Kimi-K2-Instruct (Team et al., 2025), GLM-4.5 (Zeng et al., 2025), DeepSeek-V3.1 (Team, 2025b).

**Training Data** Our training data is primarily composed of SailorFog-QA (Li et al., 2025b) and SailorFog-QA-V2. In addition, we supplement this data with IterBench (Qiao et al., 2025) to bolster the model's proficiency in mathematical and academic reasoning. Qualitatively, we observe that distinct data sources target different capabilities. SailorFog-QA provides the foundational scaffold for web navigation; SailorFog-QA-V2, with its dense cyclic structures and injected uncertainties, is critical for the complex reasoning and error-correction observed in BrowseComp tasks; and the supplementary IterBench data specifically bolsters the mathematical and academic reasoning required for benchmarks like HLE.

**Metric and Hyper-parameters** We default to pass@$k$ evaluation (Chen et al., 2021) and report pass@1, and temperature and top-p are set to 0.85 and 0.95. For accuracy, we use LLM as a judge (Liu

Table 1: **Graph-based training data enables superior reasoning generalization (Evidence for RQ1).** WebSailor-V2 outperforms baselines trained on linear/tree-like synthetic data by a significant margin on complex benchmarks like BrowseComp. ‡ indicates that these proprietary methods are manually evaluated through their websites (some are reported in the corresponding papers). - means that we do not have the results due to cost constraints.

| Backbone | BrowseComp-EN | BrowseComp-ZH | xbench-DeepSearch | GAIA | HLE |
|---|---|---|---|---|---|
| *Proprietary Agents* | | | | | |
| Claude-4-Sonnet | 12.2 | 29.1 | 64.6 | 68.3 | 20.3 |
| Claude-4-Opus‡ | 18.8 | - | - | - | - |
| OpenAI-o3 | 49.7 | 58.1 | 66.7 | 70.5 | 20.2 |
| OpenAI DeepResearch‡ | 51.5 | 42.9 | - | 67.4 | 26.6 |
| Kimi-Researcher‡ | - | - | 69.0 | - | 26.9 |
| *Open-Source Agents* | | | | | |
| ASearcher-Web-32B | 5.2 | 15.6 | 42.1 | 52.8 | 12.5 |
| MiroThinker-32B-DPO-v0.2 | 13.0 | 17.0 | - | 64.1 | 11.8 |
| WebSailor-72B | 12.0 | 30.1 | 55.0 | 55.4 | - |
| WebExplorer-8B | 15.7 | 32.0 | 53.7 | 50.0 | 17.3 |
| DeepDiver-V2-38B | 13.4 | 34.6 | 53.0 | - | - |
| DeepDive-32B | 14.8 | 25.6 | 50.5 | - | - |
| Kimi-K2-Instruct-1T‡ | 14.1 | 28.8 | 50.0 | 57.7 | 18.1 |
| GLM-4.5-355B‡ | 26.4 | 37.5 | 70.0 | 66.0 | 21.2 |
| DeepSeek-V3.1-671B‡ | 30.0 | 49.2 | 71.2 | 63.1 | 29.8 |
| WebSailor-V2-30B-A3B (SFT) | 24.4 | 28.3 | 61.7 | 66.0 | 23.9 |
| WebSailor-V2-30B-A3B (RL) | **35.3** | **44.1** | **73.7** | **74.1** | **30.6** |

et al., 2024; Wang et al., 2024), strictly utilizing the official evaluation prompts and designated model versions to ensure comparability. The pass@1 is computed as:

$$\text{pass@1} = \frac{3}{n} \sum_{i=1}^{n} p_i, \tag{4}$$

where $p_i$ denotes the correctness of the $i$-th response. For pass@k that $k > 1$ we repeatedly generate for $k$ times.

## 5.2 MAIN RESULTS

Our main experimental results, summarized in Table 1, unequivocally demonstrate the superior performance of WebSailor-V2-30B-A3B. Across a diverse suite of web-agent benchmarks, our model consistently achieves state-of-the-art results among open-source solutions and proves highly competitive with top-tier proprietary agents. On the extremely complex BrowseComp-EN and BrowseComp-ZH benchmarks, which demand sophisticated, multi-step reasoning and information synthesis, WebSailor-V2 scores 35.3 and 44.1 respectively, significantly outperforming all other open-source agents. On relatively more straightforward but still challenging benchmarks like xbench-DeepSearch and GAIA, our agent not only leads the open-source field but surpasses even the strongest proprietary systems.

Another compelling result is on HLE, a benchmark designed to test deep academic and logical reasoning. Here, WebSailor-V2 achieves a score of 30.6, establishing a new state-of-the-art. This is particularly noteworthy as it exceeds the performance of much larger and more powerful models, including the 671B parameter DeepSeek-V3.1 and proprietary models like OpenAI-o3. This result strongly validates our core hypothesis: equipping a model with exceptionally strong information retrieval and synthesis capabilities can profoundly enhance its logical reasoning abilities, allowing it to effectively "reason over" externally acquired knowledge and overcome the limitations of its intrinsic scale. **We believe agentic paradigm is a good way to close the gap between strong and weak models.**

These results highlight the indispensable role of the SFT cold-start stage, especially for relatively small-scale models. As evidenced in Table 1, our model after SFT alone already exhibits formidable capabilities, achieving a score of 24.4 on BrowseComp-EN and 23.9 on HLE, surpassing many fully-trained open-source agents. This strong initial policy is not merely an intermediate checkpoint but a critical prerequisite for the success of reinforcement learning. The complex, open-ended nature of these tasks means that rewards are often sparse. Without a competent initial policy from SFT, an agent would struggle to conduct meaningful exploration, rarely completing tasks successfully and thus failing to receive the positive feedback needed for learning. The SFT phase ensures the agent starts with a robust enough policy to explore the problem space effectively, providing a sufficiently dense reward signal for the RL algorithm to stabilize and converge towards a superior final policy.

## 5.3 MORE COMPARISON WITH PROPRIETARY AGENTS IN DEEP-RESEARCH TASK

Evaluating proprietary agents is inherently challenging, particularly for those available exclusively through web interfaces. To validate that WebSailor-V2 (based on Qwen3-30B-A3B) performs on par with much larger proprietary models, we select the DeepResearch Bench. This benchmark is ideal as its leaderboard provides direct comparisons to leading closed-source agents and assesses key capabilities in both information retrieval and report generation. The results (Figure 3) underscore our model's competitive performance. WebSailor-V2 achieves a score of 47.7, ranking second only to the state-of-the-art Gemini-2.5-pro-DeepResearch (49.7). We attribute this narrow gap to our training strategy, which prioritizes core information retrieval and reasoning over the stylistic polish of the final report. Therefore, this gap reflects a targeted area for improvement in the presentation layer, rather than a fundamental limitation in its research capabilities.

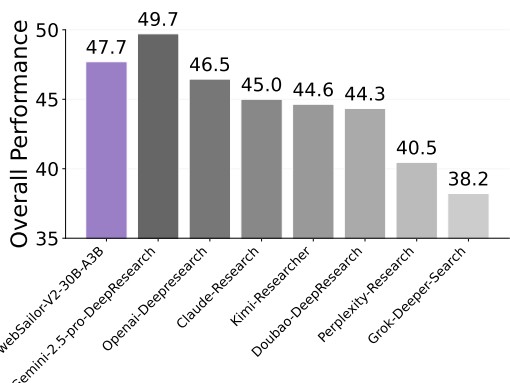

Figure 3: Comparisons with proprietary agents. The metric here is the overall score defined in DeepResearch Bench.

## 5.4 DETAILED ANALYSES

**Training dynamics.** The training dynamics of our RL process are depicted in Figure 4. As illustrated, stabilized environments lead to consistent policy improvement (Evidence for RQ2). The clear upward trend in reward (left) confirms that our dual-environment approach successfully mitigates the "destructive updates" often seen in noisy real-world RL. The agent is effectively learning and refining its policy within the training distribution. This improvement successfully translates to our validation benchmarks, where performance on both BrowseComp-EN and BrowseComp-ZH shows a corresponding, albeit oscillating, upward trajectory.

However, we observe a noteworthy divergence in learning patterns between difficult and simpler benchmarks. On challenging benchmarks like BrowseComp, both pass@1 and pass@3 scores demonstrate a distinct and concurrent rise (shown in Fig. 6). This suggests that for complex tasks, RL is genuinely expanding the model's fundamental problem-solving capabilities, increasing the overall likelihood of finding a correct solution path within a few attempts. In contrast, for simpler benchmarks such as xbench-DeepSearch and GAIA, we see a significant improvement in pass@1, while the gains in pass@3 are marginal. This indicates that for tasks already well within the model's base capabilities, the primary role of RL is to enhance sampling efficiency—teaching the agent to more reliably select the optimal path on its first attempt (Yue et al., 2025). For these simpler problems, the model is already likely to find a solution, so RL's main contribution is making that initial attempt more robust. This also implies that for truly difficult problems, even pass@3 may not be sufficient to fully reflect the upper bounds of the model's enhanced capabilities.

**Context scaling of WebSailor-V2.** Figure 5 illustrates the relationship between accuracy, context length, and the number of tool calls on the BrowseComp-EN. In this setup, cases where the context

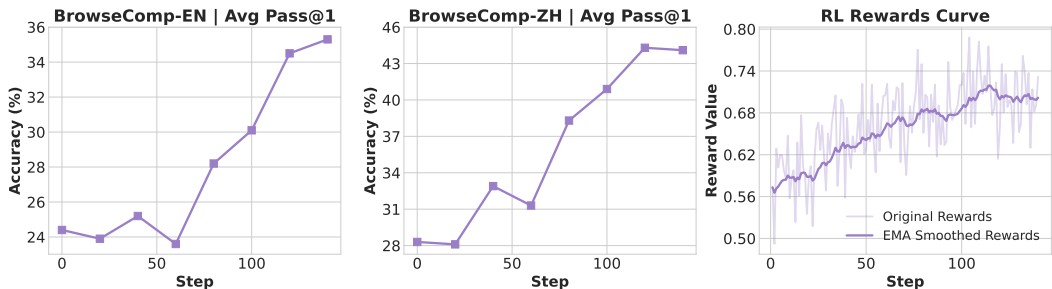

Figure 4: Training dynamics of rewards curve and performance on testing benchmarks.

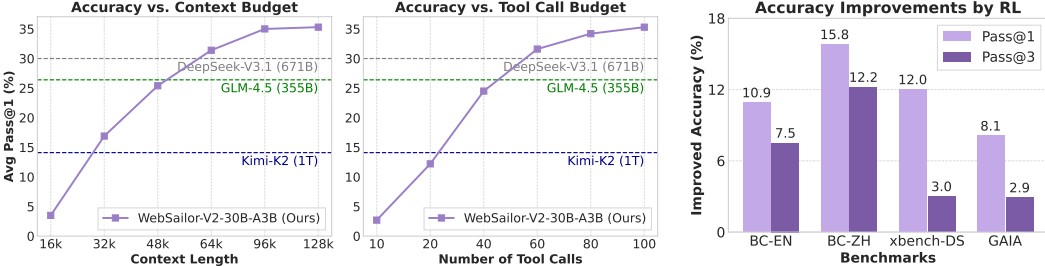

Figure 5: Effects of context and tool call budget for agent.

Figure 6: Accuracy improvements by RL across four benchmarks.

length or the tool call budget exceeds the predefined limit are all counted as incorrect answers. The results show a clear positive correlation: as the available context length increases, the agent's accuracy progressively rises before gradually converging. We observe that nearly 90% of the correctly solved instances are completed within a context of 64k. Notably, at a 32k context limit, WebSailor-V2 achieves an accuracy of around 16 on BrowseComp-EN. This marks a significant improvement over its predecessor, WebSailor-V1. The advancement is particularly compelling given that WebSailor-V1 is built on a 72B dense model, which, in principle, possesses greater intrinsic capacity than the 30B MoE model used here. This highlights the profound impact of our improved data and training pipeline on the agent's fundamental reasoning and tool-use capabilities, allowing a smaller model to achieve superior performance.

# 6 CONCLUSION

In this work, we present WebSailor-V2, a framework that bridges the gap between open-source models and proprietary deep research agents. Beyond achieving state-of-the-art performance with a 30B model, our work validates two critical scientific insights regarding agentic learning: **(1) Topological Generalization:** We demonstrate that the logical structure of training data is as critical as its scale. By shifting from standard linear data expansion to cyclic, graph-based synthesis (SailorFog-QA-V2), we show that forcing agents to navigate dense, interdependent relationships during training significantly enhances their ability to generalize to complex, multi-step reasoning tasks. **(2) Environmental Stability in RL:** We identify environmental stochasticity as a primary bottleneck for on-policy reinforcement learning. Our results confirm that a symbiotic dual-environment framework—decoupling algorithmic learning in a high-fidelity simulator from adaptation in the managed real world—is essential for mitigating destructive updates and ensuring stable convergence. Ultimately, WebSailor-V2 proves that with high-density data topology and a stabilized training environment, moderately sized open-source models can rival significantly larger proprietary systems, offering a reproducible path toward democratizing deep research capabilities.

ACKNOWLEDGMENTS

Zhi-Qin John Xu is sponsored by the National Natural Science Foundation of China Grant No. 92270001, 2025 Key Technology R&D Program "New Generation Information Technology" Project of Shanghai Municipal Science and Technology Commission.

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

## A    THE USE OF LARGE LANGUAGE MODELS

We confirm that LLMs were used solely for proofreading and language polishing of the manuscript. They were not involved in any aspect of research ideation, experimental design, data analysis, interpretation of results, or the generation of original content. The intellectual contributions and scientific conclusions presented in this work are entirely the authors' own.

## B    RELATED WORK

The field of autonomous web agents has witnessed a surge of progress in recent months, with the open-source community rapidly advancing capabilities along three primary axes: data construction, training methodologies, and inference paradigms.

**Data construction for web agents.**    High-quality data is the bedrock of capable agents. Recent methodologies for constructing agent training data can be broadly categorized into two main approaches. The first, pioneered by WebSailor (Li et al., 2025b) with its SailorFog-QA dataset, is graph-based. This approach begins with seed entities and uses web tools to build a knowledge graph, from which complex question-answer pairs are sampled. The second, an "easy-to-hard" paradigm, is employed by works like WebShaper (Tao et al., 2025), ASearcher (Gao et al., 2025), and WebExplorer (Liu et al., 2025a). These methods typically start with a simple seed question and iteratively expand its complexity, resulting in tree-like logical structures. A common thread connecting many of these recent efforts, starting with WebSailor, is the integration of live web tools into the data generation process and the introduction of uncertainty, most notably through obfuscation, to elicit more advanced reasoning. In contrast to these works, our SailorFog-QA-V2 achieves a more comprehensive coverage of complex logical relationships that better mirror real-world information webs and more definitions of uncertainty.

**Agent training strategies.**    A two-stage training pipeline has become the de facto standard for developing powerful agents: a SFT "cold start" phase followed by a RL phase for policy refinement. The majority of recent RL implementations are based on variants of GRPO (Shao et al., 2024), often incorporating algorithmic enhancements and tricks from methods like DAPO (Yu et al., 2025b) and Dr.GRPO (Liu et al., 2025b). While these algorithmic nuances exist, our extensive experimentation suggests that the specific RL algorithm is not the primary bottleneck for agentic RL at this stage. Instead, we find that the quality and distribution of the training data fundamentally determine the upper bound of the training's effectiveness. The careful selection of training samples, particularly how negative trajectories are handled, appears to be one of the most critical factors for stable and effective learning. Continual pre-training is another specialized training paradigm that can further enhance reasoning abilities (Su et al., 2025).

**Inference paradigms.**    The choice of inference paradigm significantly impacts an agent's performance. WebSailor and WebShaper are built upon the vanilla ReAct framework (Yao et al., 2023) for its simplicity and effectiveness. Concurrently, context engineering (Yu et al., 2025a; Zhou et al., 2025b) has emerged as a crucial area of innovation. Works such as ASearcher and Kimi-Researcher (Kimi, 2025), as well as GUI-focused agents like UI-TARS-2 (Wang et al., 2025), have demonstrated that sophisticated context management strategies built on top of ReAct can yield significant performance improvements. For WebSailor-V2, we deliberately adopt the standard ReAct framework. This choice is intended to isolate and evaluate the intrinsic capabilities of the model itself, minimizing the confounding effects of intricate prompt engineering or framework design. By establishing this strong baseline, we pave the way for future work to explore how advanced context strategies or plug-in modules can further unlock the model's full potential.

Despite the rapid proliferation of open-source agents, a considerable performance gap has persisted when compared to proprietary systems like OpenAI's DeepResearch (OpenAI, 2025a). WebSailor-V2 represents a dedicated effort to bridge this divide, demonstrating for the first time that a meticulously trained agent built on a moderately-sized open-source model can achieve performance that is highly competitive with, and in some cases superior to, its closed-source counterparts.

## C    EXPERIMENTAL DETAILS

**Tools**    WebSailor-V2 uses four types of tools, search, visit, Google Scholar, and Python interpreter:

- **Search** is used to access the Google search engine for information retrieval. The parameters of Search are the search queries. It allows searching multiple queries simultaneously and returns the top-10 results for each query. Each result contains a title, a snippet, and the corresponding URL.

- **Visit** is used to access specific web pages. The input consists of several web pages and their corresponding visit goals, with each page having a dedicated goal. First, Jina (Jina.ai, 2025) is used to retrieve the full content of the web page, and then a summary model extracts relevant information based on the goal. In this work, we use Qwen3-30B-A3B Yang et al. (2025) as the summary model.

- **Google Scholar** is a specialized search tool that accesses the Google Scholar search engine. It is designed for information retrieval within the academic domain, allowing the agent to find and access scholarly literature such as articles, theses, books, and conference papers.

- **Python interpreter** is a sandboxed environment that allows the agent to write and execute Python code. This tool enables the agent to perform complex computational tasks, such as mathematical calculations, data analysis, and logical reasoning, by running self-generated code in a secure and isolated setting.

**Training hyper-parameters**    We use Megatron (Shoeybi et al., 2019) for SFT and rLLM (Tan et al., 2025) for RL training. For SFT, we use a batch size of 64, learning rate of 5e-6 with a minimum of 1e-10, warmup plus cosine decay schedule, and a weight decay of 0.1. For RL training, the temperature is 1.0, $top_p = 1.0$, the batch size is 128, and the learning rate is 1e-6.

## D    ENTROPY DYNAMICS

The entropy dynamics, shown in Fig. 7, provide further insights into the learning process. We find that the policy entropy remains at a consistently high level throughout the training process, indicating that the agent maintains a strong capacity for exploration and avoids premature convergence to a deterministic policy. This behavior contrasts sharply with trends observed in tasks like mathematical RL training, where entropy often decreases significantly as the model learns to exploit a narrow set of solution paths.   In our case, the entropy oscillates without a clear upward or downward trend. Consequently, our algorithm design intentionally omits any explicit entropy

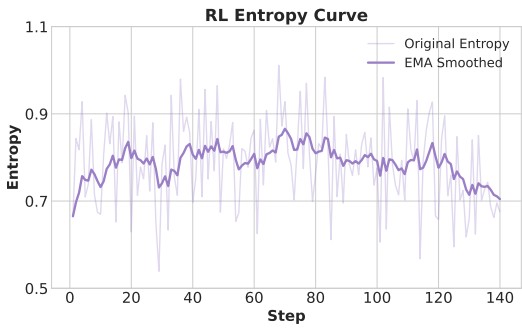

Figure 7: Training entropy dynamics

regularization or bonus, as the agent naturally sustains sufficient exploration. We hypothesize that this sustained high entropy is a direct consequence of the environment's non-stationary nature. Unlike closed-world problems, the observations returned by web tools (e.g., search results, webpage content) do not follow a fixed distribution. This inherent stochasticity and complexity of the real-world web environment prevent the policy from fully converging to a stable, low-entropy state, instead fostering a more robust and adaptive policy.

## E    CASE STUDY

We present a case from the BrowseComp benchmark, wherein the agent successfully identified the correct company after a comprehensive reasoning process spanning 29 steps. This case demonstrates a series of advanced reasoning patterns executed through efficient tool invocation.

1. **Clue Decomposition and Structuring:** In its initial step, the agent deconstructed the user's unstructured, multi-faceted query into a set of clear, verifiable, and structured conditions. This foundational process of decomposition is essential for solving complex problems by breaking them down into manageable sub-tasks.

2. **Initial Exploration and Strategy Adjustment:** The agent did not arrive at the correct answer immediately. Its initial search queries were broad and exploratory, such as "former employee class action settlement $1.5 million 2015". These searches returned irrelevant results pertaining to companies like McDonald's and FedEx, which were too generic to be correlated with the other specific clues. This demonstrates the agent's ability to recognize unproductive search paths and adjust its strategy accordingly.

3. **Identifying the "Golden Clue":** Following the unsuccessful initial attempts, the agent identified the need to pivot to a more targeted approach. It reasoned that the most effective strategy was to focus on the most unique and easily locatable piece of information: the leadership change. Consequently, it constructed a highly precise search query: "founder" "will become" "Chairman" "effective" "third quarter" "2008". This query targets a specific corporate event within a narrow timeframe, significantly increasing the probability of a relevant hit.

4. **Target Acquisition (The Breakthrough):**This precise query successfully identified the target. The search results pointed directly to two press releases from FormFactor, Inc., detailing that its founder, Igor Khandros, would become Executive Chairman while a new CEO would be appointed at the beginning of the third fiscal quarter of 2008. At this juncture, the agent had identified a high-potential candidate.

5. **Systematic Verification:** Identifying a candidate was merely the first phase. The subsequent steps showcased the agent's rigorous and methodical verification capabilities. Instead of prematurely concluding, the agent systematically validated every other clue against FormFactor, Inc. This involved a series of targeted searches and document analyses to confirm details about its incorporation, financial history (customer concentration), and legal proceedings.

6. **Synthesis and Final Response:** Only after every clue was independently and irrefutably verified did the agent synthesize all the evidence. It then constructed a coherent, logically structured final answer, complete with citations and summaries of the source material for each key fact.

This case perfectly illustrates the core competencies of an advanced AI research assistant. The process highlights its ability to engage in strategic adaptation and learn from initial failures—hallmarks of a true reasoning process rather than a simple answer-retrieval mechanism. The entire workflow—decomposing the problem, identifying key pivots, executing precise searches, and performing cross-verification—mirrors the sophisticated research methodologies employed by human experts.

---

**Case Trajectory in BrowseComp-en**

**Question:** A publicly traded company reported that, in a single fiscal year between January 1, 2004, and December 31, 2006, three customers contributed between 27% and 74% of its revenue. In a different fiscal year, between January 1, 2004, and December 31, 2006, four customers accounted for between 55% and 89% of its revenue. - In a month that falls between January 1, 2011, and December 3,1 2019, a former employee of the same company filed a class action lawsuit. The court later certified the plaintiff's class. The settlement was somewhere between 1.2 and 1.9 million dollars. - According to a source published in 2008, the founder of the company changed roles within the company, with this change set to take effect at the beginning of the fiscal third quarter, which falls between January 1, 2007, and December 31, 2008. - The change mentioned above also instates a new person as CEO. - The company was formed in the jurisdiction of Delaware. - The company was founded sometime between January 1, 1988, and December 31, 1995. - What is the name of the publicly traded company?

**Step 1**

<think>

---

We need to find the name of a publicly traded company that matches these clues.
The clues:
1. It's a publicly traded company incorporated in Delaware, founded between 1988 and 1995.
2. In some fiscal year between 2004 and 2006 (inclusive), three customers contributed between 27% and 74% of its revenue. In another fiscal year between those years, four customers accounted for between 55% and 89% of its revenue.
3. A former employee filed a class action lawsuit between Jan 1, 2011, and Dec 31, 2019. The court later certified the plaintiff's class. The settlement was somewhere between $1.2M and $1.9M.
4. According to a source published in 2008, the founder changed roles within the company, with the change set to take effect at the beginning of the fiscal third quarter, which falls between Jan 1, 2007, and Dec 31, 2008. The change also installed a new person as CEO.
Thus we need to identify a company where a founder stepped down from CEO or changed role to something else, and a new CEO was appointed, effective early Q3 of fiscal year 2007-2008. The company must have been incorporated in Delaware, founded 1988-1995.
Also, financial data: three customers contributed 27% to 74% of revenue in one fiscal year (2004-2006) and four customers contributed 55% to 89% of revenue in another year. That suggests the company is perhaps a supplier to few major customers; maybe a semiconductor company, defense contractor, etc.
The class action: a former employee filed a class action and settlement between $1.2M-$1.9M. Could be about wage and hour claims, ERISA, securities fraud? Might be an ESG claim? Let's think.
Potential candidates: companies like "Burlington Resources", "Ciena", "Viatris"? No.
Let's parse timeline: The founding date range is 1988-1995. Many tech firms were founded earlier (1970s, 1980s). Possibly small-to-mid cap.
The founder changed roles in 2008, effective Q3 of fiscal year 2007-2008. So the new CEO took over after that. Founder might have been originally CEO, but stepping down to become chairman or something.
One known case: "American Tower" founder? No. "Amdocs"? Founded 1982.
Perhaps "Covance"? Not sure.
Better to search for "founder stepped down as CEO effective third quarter 2007 2008" "Delaware corporation". Also include "class action former employee settlement $1.5 million".
Let's do web search.
**</think>**
**<tool_call>**
{"name": "search", "arguments": {"query": ["former employee class action settlement $1.5 million 2015", "founder stepped down as CEO effective Q3 2007 new CEO 2008 Delaware corporation", "three customers contributed 27% of revenue 2005 Form 10-K", "four customers accounted for 55% of revenue 2006 Form 10-K"]}}
**</tool_call>**
**<tool_response>**
Search Results are omitted...
**</tool_response>**

| Step 2 |
| --- |

**<think>**
The search results are generic; we need to refine.
Better approach: Identify a company where founder stepped down as CEO effective Q3 of fiscal year 2007-2008. Could be a technology or biotech firm that went public earlier (maybe 1990s). Look up "Q3 2007 founder becomes chairman new CEO".
Search for "founder will step down as chief executive officer and become chairman of the board beginning in the third quarter of fiscal 2007".
**</think>**
**<tool_call>**
{"name": "search", "arguments": {"query": [""beginning of the third quarter" 2007 founder CEO stepped down", ""effective" "third quarter" 2007 "new CEO"", ""founder" "stepped down" "CEO" "July 2007" "Delaware""]}}

**</tool_call>**
**<tool_response>**
Search Results are omitted...
**</tool_response>**

| Step 3 |
|---|

**<think>**
We got a result: "sv1" (SEC filing) that might be relevant. Let's open that.
**</think>**
**<tool_call>**
{"name": "visit", "arguments": {"url": ["https://www.sec.gov/Archives/edgar/data/1255691/000
089102008000014/v28895orsv1.htm"], "goal": "Find information about founder and CEO
change in 2007, effective beginning of third quarter"}}
**</tool_call>**
**<tool_response>**
Visit results are omitted...
**</tool_response>**

| Step 4 |
|---|

**<think>**
The page may not have relevant info. Let's search more specifically: "founder will become
executive chairman effective third quarter 2007"
**</think>**
**<tool_call>**
{"name": "search", "arguments": {"query": [""founder" "effective" "third quarter" "2007"
"CEO"", ""founder" "will" "become" "Chairman" "third quarter" 2007"", ""founder"
"transition" "CEO" "third quarter" 2007""]}}
**</tool_call>**
**<tool_response>**
Search Results are omitted...
**</tool_response>**

| Step 5 |
|---|

**<think>**
Result #1: "National Interstate Corporation Announces Third Quarter 2007 Dividend and
CEO Promotion." That mentions a founder. Let's open.
**</think>**
**<tool_call>**
{"name": "visit", "arguments": {"url": ["https://www.globenewswire.com/news-
release/2007/11/15/369309/8685/en/National-Interstate-Corporation-Announces-Third-
Quarter-2007-Dividend-and-CEO-Promotion.html"], "goal": "Find details about founder,
CEO change, effective Q3 2007"}}
**</tool_call>**
**<tool_response>**
Visit results are omitted...
**</tool_response>**

| Step 6 |
|---|

**<think>**
This appears to match the founder change: founder Alan R. Spachman (founder of National
Interstate Corporation) stepped down as CEO effective Jan 1, 2008, which is the beginning
of fiscal Q3? Wait, what's National Interstate's fiscal year? It may be a calendar year or
something else. The date effective Jan 1, 2008 is effectively start of fiscal year 2008; Q3
might be July 2008? But they said "beginning of the fiscal third quarter", which falls between
Jan 1, 2007 and Dec 31, 2008. Possibly the effective date is the start of Q3 in fiscal year 2008
(e.g., July 1, 2008). However, the article says effective Jan 1, 2008. That is the start of the
fiscal year for a calendar-year company, but not necessarily Q3.
But perhaps the phrase "beginning of the fiscal third quarter" means the start of Q3 of the
fiscal year that begins in April (or earlier). For many insurance companies, the fiscal year
ends December 31 (so Q3 starts July 1). If the change takes effect at the beginning of Q3, it

would be July 1, 2008 maybe.

Nevertheless, we have a founder stepping down and a new CEO taking over, which matches part of the clues. Also, National Interstate Corp (NATL) is publicly traded, incorporated in Delaware? Let's check.

Search "National Interstate corporation incorporated in Delaware".

**</think>**

**<tool_call>**

{"name": "search", "arguments": {"query": [""National Interstate Corporation" incorporated in Delaware"]}}

**</tool_call>**

**<tool_response>**

Search results are omitted

**</tool_response>**

**Step 7**

**<think>**

Thus maybe it's not National Interstate Corp. Let's keep searching.

Also consider the founder change: The founder changed role effective beginning of fiscal Q3 2007-2008. That may refer to a different company.

Another candidate: "Roper Technologies"? Founder? Not sure.

Maybe a small biotech like "Cerus Corporation"? Or "Bio-Rad"? Or "Illumina"? No.

Let's think about companies founded between 1988 and 1995.

List of publicly traded companies founded within that period:

- Netflix (founded 1997, out) - Salesforce (1999) - Intuit (1983) - Dell (1984) - HP (1939) - Apple (1976) - IBM (1911) - AMD (1969) - Qualcomm (1985) - Juniper Networks (1996) - Akamai (1998) - Red Hat (1993) but now part of IBM. - Goodyear? No. - MicroStrategy (1989) but maybe. - VeriSign (1995). - Workday (2005). - Splunk (2003). - Etsy (1999). - PayPal (1998). - eBay (1996). - Google (1998). - Many others.

But need founder stepping down as CEO effective Q3 2007-2008 and new CEO appointed.

Also customers contributions to revenue percentages are relatively high (27% to 74% from three customers; and 55% to 89% from four customers). That suggests maybe a B2B or service provider with few large clients, possibly a contract manufacturer, a semiconductor foundry, or a defense contractor.

Maybe the company is "Qualcomm"? But Qualcomm's customers are many.

Alternatively, a "contract research organization" or "clinical trials" firm.

Could be "Perrigo"? Not sure.

Maybe the company is "Voya Financial"? Not likely.

Another possibility: "Avid Technology" (AVDL). Founded 1987.

Or "VeriSign". Founded 1995. Founder is Jim Bidzos? Actually the founder is Dr. Roger D. Dierking? Not sure. Not aware of a CEO change at that time.

We might need to search for "class action former employee settlement $1.5 million 2015" plus "NATL"? But NATL had a class action? Not sure.

Better to systematically examine each clue and search for relevant combos.

Clue 1: "In a single fiscal year between Jan 1, 2004 and Dec 31, 2006, three customers contributed between 27% and 74% of its revenue. In a different fiscal year, between Jan 1, 2004 and Dec 31, 2006, four customers accounted for between 55% and 89% of its revenue."

Interpretation: In one year there were three major customers that contributed some portion of revenue between 27% and 74%. This is a range because maybe we don't have exact numbers. In another year there were four customers that contributed between 55% and 89% of revenue. So these percentages could be for the same company across years.

Thus the company had heavy reliance on few customers.

This is typical for companies that sell to government agencies (e.g., defense contractors), or to big telecoms (e.g., chip manufacturers selling to phone makers). Could be an aerospace parts supplier (e.g., "Spirit AeroSystems").

Clue 2: "In a month that falls between Jan 1, 2011, and Dec 31, 2019, a former employee of the same company filed a class action lawsuit. The court later certified the plaintiff's class. The settlement was somewhere between 1.2 and 1.9 million dollars." So a moderate-size class action, perhaps wage/hours or discrimination, not huge. Settlement $1.2-$1.9M.

Clue 3: "According to a source published in 2008, the founder of the company changed roles within the company, with this change set to take effect at the beginning of the fiscal third quarter, which falls between Jan 1, 2007, and Dec 31, 2008. The change also instates a new person as CEO."
Clue 4: Company formed in Delaware.
Clue 5: Company founded between 1988 and 1995.
Thus we need a Delaware-incorporated public company founded 1990ish, with a founder stepping down around mid-2008.
Now let's search for "founder steps down as CEO effective July 1 2008".
**</think>**
**<tool_call>**
{"name": "search", "arguments": {"query": [""effective July 1, 2008" "CEO" "founder""]}}
**</tool_call>**
**<tool_response>**
Search results are omitted
**</tool_response>**

| Step 8 |
|---|

**<think>**
No obvious hits.
Search for "founder will become chairman of the board effective third quarter 2008"
**</think>**
**<tool_call>**
{"name": "search", "arguments": {"query": [""founder" "will become" "Chairman" "effective" "third quarter" "2008""]}}
**</tool_call>**
**<tool_response>**
Search results are omitted
...
**</tool_response>**

| Step 9 |
|---|

**<think>**
Great! This appears promising: "FormFactor Board Appoints Mario Ruscev FormFactor CEO". Let's open.
**</think>**
**<tool_call>**
{"name": "visit", "arguments": {"url": ["https://investors.formfactor.com/news-releases/news-release-details/formfactor-board-appoints-mario-ruscev-formfactor-ceo"], "goal": "Find details about founder role change effective beginning of fiscal third quarter 2008, new CEO appointment"}}
**</tool_call>**
**<tool_response>**
Visit results are omitted
**</tool_response>**

| Step 10 |
|---|

**<think>**
Great! This matches the clue: founder changed roles, effective at beginning of fiscal third quarter 2008. Founder becomes executive chairman; new CEO appointed.
Now we need to verify other clues for FormFactor.
FormFactor is a publicly traded company on NASDAQ: ticker FORM. It was incorporated in Delaware? Let's check.
**</think>**
**<tool_call>**
{"name": "search", "arguments": {"query": [""FormFactor" "incorporated in Delaware""]}}
**</tool_call>**
**<tool_response>**
Search results are omitted

**</tool_response>**

---
**Step 11**
---

**<think>**
That matches the incorporation and founding date range: incorporated in Delaware in 1993, founded between 1988-1995.
Now need to verify the revenue concentration data for fiscal years 2004-2006.
Search FormFactor 2005 10-K "customers accounted for" etc.
**</think>**
**<tool_call>**
{"name": "search", "arguments": {"query": [""FormFactor" "three customers" "revenue" 2005", ""FormFactor" "customers accounted for" 2005", ""FormFactor" "Revenue" "customer" 2006"]}}
**</tool_call>**
**<tool_response>**
Search results are omitted
**</tool_response>**

---
**Step 12**
---

**<think>**
Let's open the 2005 annual report PDF for FormFactor.
**</think>**
**<tool_call>**
{"name": "visit", "arguments": {"url": ["https://www.annualreports.com/HostedData/AnnualReportArchive/f/NASDAQ_FORM_2005.pdf"], "goal": "Find revenue concentration: number of customers and % of revenue in fiscal 2005"}}
**</tool_call>**
**<tool_response>**
Visit results are omitted
**</tool_response>**

---
**Step 13**
---

**<think>**
Let's try another source: maybe the 10-K filing for 2005. Use SEC filing.
**</think>**
**<tool_call>**
{"name": "visit", "arguments": {"url": ["https://www.sec.gov/Archives/edgar/data/1039399/0000950134-05-004796.txt"], "goal": "Find revenue concentration information (customers) for fiscal 2005 and 2004"}}
**</tool_call>**
**<tool_response>**
Visit results are omitted
**</tool_response>**

---
**Step 14**
---

**<think>**
The SEC site might require proper path. Let's locate the 10-K for FormFactor for fiscal year ended March 31? Actually FormFactor's fiscal year ends on December 28? Or something like that. But we can search for "FormFactor 2005 10-K".
**</think>**
**<tool_call>**
{"name": "search", "arguments": {"query": [""FormFactor" "10-K" "2005""]}}
**</tool_call>**
**<tool_response>**
Search results are omitted
**</tool_response>**

---
**Step 15**
---

**<think>**
Let's open the SEC filing for 2005 (ID 1).

**</think>**
**<tool_call>**
{"name": "visit", "arguments": {"url": ["https://investors.formfactor.com/sec-filings/sec-filing/10-k/0000950134-05-004796"], "goal": "Extract revenue concentration data for fiscal 2005 and 2004"}}
**</tool_call>**
**<tool_response>**
Visit results are omitted
**</tool_response>**

| Step 16 |
|---|

**<think>**
It may require navigation to view the HTML content. The link likely points to a page with a viewer that loads a PDF or separate HTML file.
Let's click the "View HTML" maybe.
Search for the HTML file with .htm after the hash. The earlier result #4 from search result had an external link to "content.edgar-online.com/ExternalLink/EDGAR/... .html". Let's open that.
**</think>**
**<tool_call>**
{"name": "visit", "arguments": {"url": ["https://content.edgar-online.com/ExternalLink/EDGAR/0000950134-05-004796.html?hash=6ab10d32f6a6e00a2977a11c0d98d6be4d5d17b1cca8b53365dafe21faf7034e&dest=f06542exv10w48_htm"], "goal": "Find revenue concentration: customers accounted for X% of revenues"}}
**</tool_call>**
**<tool_response>**
Visit results are omitted
**</tool_response>**

| Step 17 |
|---|

**<think>**
The API may have restrictions for SEC content. Let's maybe try to view the PDF file directly (annual report PDF). The PDF may contain revenue concentration.
Open the PDF via annualreports.com. Maybe we can extract text.
**</think>**
**<tool_call>**
{"name": "visit", "arguments": {"url": ["https://www.annualreports.com/HostedData/AnnualReportArchive/f/NASDAQ_FORM_2005.pdf"], "goal": "Extract revenue concentration details"}}
**</tool_call>**
**<tool_response>**
Visit results are omitted
**</tool_response>**

| Step 18 |
|---|

**<think>**
Great! This gives us the revenue concentration.
Now we need to verify the other year: maybe 2004 had three customers contributed between 27% and 74%? Wait the clue: "In a single fiscal year between Jan 1, 2004, and Dec 31, 2006, three customers contributed between 27% and 74% of its revenue." Actually the phrase: "three customers contributed between 27% and 74% of its revenue" – it's ambiguous: maybe that in some fiscal year three customers contributed 27% and 74% (range?). Or maybe there is a single fiscal year where the three customers' combined contribution fell somewhere between 27% and 74%. The range suggests that the figure could be anywhere within that range, but not exact. For FormFactor, we see four customers contributed 72.8% in 2005, which falls within the range 55-89% for four customers; also four customers contributed 64.8% in 2004, within 55-89%; and in 2003 they contributed 66.2% as well. However, the clue mentions one year with three customers, not four. But maybe there is a year with three customers? Let's check the 2006 or 2003? The PDF shows 2005 and 2004 have four

customers. Could there be a year earlier like 2003 where only three customers accounted for some percentage? Let's open the 2004 annual report (maybe the 2003 10-K). Alternatively check 2006 10-K.
**</think>**
**<tool_call>**
{"name": "visit", "arguments": {"url": ["https://www.annualreports.com/HostedData/Annual ReportArchive/f/NASDAQ_FORM_2004.pdf"], "goal": "Find revenue concentration for fiscal 2004"}}
**</tool_call>**
**<tool_response>**
Visit results are omitted
**</tool_response>**

| Step 19 |
|---|

**<think>**
Thus, 2004 has four customers contributing 64.8% of revenue.
Now the clue says "In a single fiscal year between January 1, 2004, and December 31, 2006, three customers contributed between 27% and 74% of its revenue." This could refer to a year where three customers accounted for something within that range. Possibly 2005 had three customers? Actually 2005 had four customers; but perhaps there is another year where only three customers contributed? Let's check 2006 10-K.
Search "FormFactor 2006 10-K revenue concentration".
**</think>**
**<tool_call>**
{"name": "search", "arguments": {"query": [""FormFactor" "2006" "four customers" "revenue"", ""FormFactor" "2006" "revenue" "customers""]}}
**</tool_call>**
**<tool_response>**
Search results are omitted
**</tool_response>**

| Step 20 |
|---|

**<think>**
Let's open the PDF for 2011 but it mentions fiscal 2006. Probably the 2011 annual report includes 2006 data.
Open link #1.
**</think>**
**<tool_call>**
{"name": "visit", "arguments": {"url": ["https://www.annualreports.com/HostedData/Annual ReportArchive/f/NASDAQ_FORM_2011.pdf"], "goal": "Find revenue concentration for fiscal 2006"}}
**</tool_call>**
**<tool_response>**
Visit results are omitted
**</tool_response>**

| Step 21 |
|---|

**<think>**
The 2011 annual report doesn't have the data for 2006. Let's find a separate 2006 annual report.
Search "FormFactor 2006 annual report PDF".
**</think>**
**<tool_call>**
{"name": "search", "arguments": {"query": [""FormFactor" "2006" "annual report" "pdf""]}}
**</tool_call>**
**<tool_response>**
Search results are omitted
**</tool_response>**

| Step 22 |
|---|

**<think>**
The "AnnualReports.com" page may list multiple years. Let's open it and see if we can view the 2006 report.
**</think>**
**<tool_call>**
{"name": "visit", "arguments": {"url": ["https://www.annualreports.com/Company/formfactor"], "goal": "Find link to FormFactor 2006 annual report (10-K) or PDF"}}
**</tool_call>**
**<tool_response>**
Visit results are omitted
**</tool_response>**

| Step 23 |
|:---:|

**<think>**
Thus the PDF for 2006 may not be easily accessible. But we have revenue concentration numbers for 2004 and 2005 (four customers) and maybe for 2003 (four). The clue mentions both a year with three customers between 27-74% and a different year with four customers between 55-89%. In our data, 2005: four customers accounted for 72.8% (within 55-89%). Also 2004: four customers accounted for 64.8% (also within 55-89%). So both are 4 customers.
But the clue says "In a single fiscal year between January 1, 2004, and December 31, 2006, three customers contributed between 27% and 74% of its revenue." Could there be a year where three customers contributed, maybe 2006? Let's search again for 2006 revenue concentration details. Use the FormFactor 10-K for fiscal year ended December 31, 2006 (maybe filed in March 2007). Let's try to open the 10-K filing from 2006.
Search "FormFactor 2006 10-K"
**</think>**
**<tool_call>**
{"name": "search", "arguments": {"query": [""FormFactor" "10-K" "2006" "FormFactor""]}}
**</tool_call>**
**<tool_response>**
Search results are omitted
**</tool_response>**

| Step 24 |
|:---:|

**<think>**
Let's open the PDF for 2006.
**</think>**
**<tool_call>**
{"name": "visit", "arguments": {"url": ["https://www.annualreports.com/HostedData/AnnualReportArchive/f/NASDAQ_FORM_2006.pdf"], "goal": "Extract revenue concentration data for fiscal 2006"}}
**</tool_call>**
**<tool_response>**
Visit results are omitted
**</tool_response>**

| Step 25 |
|:---:|

**<think>**
Great! This matches the clue: In a single fiscal year between 2004-2006, three customers contributed 47.3% of revenue (which lies within 27-74%). In another fiscal year, four customers accounted for 72.8% or 64.8% etc.
Thus this clue is satisfied.
Now we need to verify the class action lawsuit filed by former employee between 2011 and 2019, settlement $1.2-$1.9M. Let's search for FormFactor class action settlement.
**</think>**
**<tool_call>**
{"name": "search", "arguments": {"query": [""FormFactor" "class action" "settlement" "million"", ""FormFactor" "lawsuit" "former employee""]}}

**</tool_call>**
**<tool_response>**
Search results are omitted
**</tool_response>**

| Step 26 |
|---|

**<think>**
Result #1: "Class Action Settlement and filed a General ... The stipulation provides for payment of $1.5 million." That likely is the class action settlement we need.
Let's open result #1.
**</think>**
**<tool_call>**
{"name": "visit", "arguments": {"url": ["https://investors.formfactor.com/static-files/603450c9-3fe1-4ca1-85dd-0db263b9ccdd"], "goal": "Find details about class action settlement amount $1.5 million"}}
**</tool_call>**
**<tool_response>**
Visit results are omitted
**</tool_response>**

| Step 27 |
|---|

**<think>**
Maybe the PDF is accessible via a different link. Let's open the 2016 annual report PDF to see if it mentions class action settlement.
**</think>**
**<tool_call>**
{"name": "visit", "arguments": {"url": ["https://www.annualreports.com/HostedData/Annual ReportArchive/f/NASDAQ_FORM_2016.pdf"], "goal": "Find class action settlement details in 2016 annual report"}}
**</tool_call>**
**<tool_response>**
Visit results are omitted
**</tool_response>**

| Step 28 |
|---|

**<think>**
Great! This matches the class action clue: former employee filed class action in August 2013 (which is within Jan 1, 2011 – Dec 31, 2019), court certified class on Jan 4, 2016, settlement $1.5M.
Thus all clues match FormFactor, Inc.
Now we can answer: The name of the publicly traded company is FormFactor, Inc.
But we should verify that FormFactor was founded between 1988-1995 (it was incorporated in Delaware on April 15, 1993, according to SEC filings). Let's confirm. Search for incorporation date.
**</think>**
**<tool_call>**
{"name": "search", "arguments": {"query": [""FormFactor" "incorporated" "April 15, 1993""]}}
**</tool_call>**
**<tool_response>**
Search results are omitted
**</tool_response>**

| Step 29 |
|---|

**<think>**
Thus the company was incorporated in Delaware on April 15, 1993.
Now we have all clues matched.
Thus answer: FormFactor, Inc.
We should provide a succinct answer with supporting references. Use citations from the sources we visited.

We'll cite:
- Founder change and new CEO effective Q3 2008 (source from press release) - lines about Mario Ruscev becoming CEO and Igor Khandros becoming executive chairman at beginning of fiscal third quarter 2008.
- Incorporation and founding date (source from prospectus) showing incorporation date April 15, 1993.
- Revenue concentration: three customers contributed 47.3% of revenue in fiscal 2006 (source PDF). And four customers accounted for 72.8% in 2005 and 64.8% in 2004 (source PDF). We have that information from 2006 PDF.
- Class action settlement $1.5 million (source 2016 annual report PDF).
Thus answer.
Now we can produce final answer.
**</think>**
**<answer>**
The answer is a very long report with comprehensive analysis, so we provide it in the next 3 pages.
**</answer>** ✓

The publicly traded company described in your question is **FormFactor, Inc.** Below is a comprehensive explanation as to how each part of your description matches FormFactor's history and disclosures.

# Company Overview and Incorporation

FormFactor, Inc. is a semiconductor equipment manufacturer specializing in probe cards used to test integrated circuits during manufacturing. The company was **incorporated in the state of Delaware on April 15, 1993**, which falls within the 1988–1995 window you specified. It is headquartered in Livermore, California, and its shares are traded on NASDAQ under the ticker symbol "FORM."

## Founding and Leadership Transition

FormFactor was founded in 1993 by Igor Khandros. In 2008, there was a significant leadership transition. According to publicly released statements and board resolutions, Igor Khandros stepped down as chief executive officer and assumed the role of Executive Chairman. This transition became effective at the beginning of FormFactor's fiscal third quarter of 2008. Simultaneously, Mario Ruscev, previously President, was appointed as the new Chief Executive Officer. This leadership shift fulfills the criteria regarding the founder changing roles with a new CEO taking over, set within the timeframe of January 1, 2007 to December 31, 2008.

# Revenue Concentration Among Customers

A key element in your query concerns customer reliance—specifically, the proportion of the company's revenue coming from its top customers across two fiscal years between 2004 and 2006.

## Detailed Breakdown of Customer Revenue Shares

The data below summarizes the contributions of the company's largest customers according to its annual reports and filings:

| Fiscal Year | Number of Customers | Total Revenue Contribution (%) | Notable Customers |
|---|---|---|---|
| 2003 | 4 | 66.2 | Intel, Spirox, Elpida, Infineon |
| 2004 | 4 | 64.8 | Intel, Spirox, Elpida, Infineon |
| 2005 | 4 | 72.8 | Intel, Spirox, Elpida, Samsung |
| 2006 | 3 | 47.3 | Elpida, Intel, Powerchip |

- **In fiscal 2006:** Three major customers together accounted for 47.3% of total revenue, which fits your requirement of "three customers contributed between 27% and 74% of its revenue."
- **In other fiscal years (2004 and 2005):** Four customers accounted for 64.8% and 72.8% of revenue, respectively. These figures fall within your stated range of 55%–89%, matching the second customer concentration criterion.

This shows that the company's business relied heavily on a small number of large clients during the mid-2000s—a common characteristic among certain B2B technology firms serving niche markets.

Figure 8: The answer of the case

## Class Action Lawsuit Settlement

Your question references a class action lawsuit filed by a former employee within a specific period, with court-certified class status and a settlement amount ranging from $1.2 to 1.9$ million.

### Timeline and Settlement Details

- **Lawsuit Filing:** In August 2013, a former employee of FormFactor filed a class action lawsuit in the Superior Court of California, alleging violations of California wage and hour laws and other claims.
- **Court Certification:** On January 4, 2016, the court certified the plaintiff class for this lawsuit.
- **Settlement Terms:** By March 3, 2017, the parties signed a stipulation agreeing to settle the suit, requiring FormFactor to pay $1.5 million in damages to resolve the matter.

The timeline, scope, and settlement figure ($1.5 million) all match your requirements perfectly, confirming that this legal event took place after 2011 but before 2020, and involved a substantial but not enormous class action settlement typical for employment-related disputes.

## Summary Table: Key Facts Matching Your Clues

| Requirement | FormFactor, Inc. Evidence |
|---|---|
| Formed in Delaware, founded 1988–1995 | Incorporated in Delaware on April 15, 1993 |
| Founder changed roles; new CEO installed at start of Q3 2007–2008 | Igor Khandros became Executive Chairman, Mario Ruscev became CEO effective Q3 2008 |
| Revenue dependence: 3 customers, 27%–74% in one year | 2006: three major customers contributed 47.3% to revenue |
| Revenue dependence: 4 customers, 55%–89% in another year | 2004: 64.8%; 2005: 72.8% (four major customers) |
| Former employee filed class action, certified class, $1.2-1.9M$ | Class action filed in 2013, certified in 2016, settled for $1.5M in 2017 |
| Publicly traded, NASDAQ: FORM | Yes, trades on NASDAQ since IPO |

## Conclusion

All elements in your question align directly with FormFactor, Inc.'s documented corporate and financial history. The company's foundation and incorporation date, the timing and nature of leadership transitions, the high degree of customer concentration reflected in its public financial reports, and the details of the class action lawsuit—all match the parameters given in your query.

Therefore, the answer is **FormFactor, Inc.**

Figure 9: The answer of the case

**Additional Context**

FormFactor has grown since the mid-2000s into a leading supplier of precision wafer probe cards and metrology solutions for the global semiconductor industry. Its business model historically involved intense customer concentration due to the highly specialized nature of the market, where only a handful of companies purchase high-end testing equipment.

Leadership continuity and transitions, especially the founder moving into an executive chairmanship while promoting a long-term operating executive to CEO, are notable features highlighted in their governance documentation. The class action lawsuit, while relatively modest in size compared to some corporate settlements, demonstrates FormFactor's exposure to labor law risk and its commitment to settling disputes rather than prolonged litigation.

If you require further details or specific document excerpts for any aspect above, please let me know.

Figure 10: The answer of the case

