# OpenReview forum: "WebSailor-V2: Bridging the Chasm to Proprietary Agents via Synthetic Data and Scalable Reinforcement Learning"
_ICLR.cc/2026/Conference — ICLR 2026 Poster_

### Official Review · Reviewer_uNmb · 2025-10-29

**Soundness:** 1
**Presentation:** 2
**Contribution:** 1
**Rating:** 2
**Confidence:** 4

**Summary:**

The paper presents WebSailor-V2, a post-training framework for pre-trained LLMs designed to develop agentic capabilities by integrating reasoning and tool use.
The pipeline comprises two stages: supervised fine-tuning and reinforcement learning.
Training data are derived from a densely constructed knowledge graph, from which a novel QA dataset is sampled.
For RL training, the authors develop a simulation environment that enables efficient experimentation by approximating real-world interactions, which are inherently slow and fragile.
The final training stage is executed in real-world conditions using a custom tool-calling API.
Empirical results show that WebSailor-V2 surpasses both state-of-the-art open-source and most proprietary models on benchmarks assessing agentic reasoning.
However, the paper provides insufficient methodological details, limiting reproducibility and interpretability of the proposed techniques.
Furthermore, many claims are presented without supporting evaluations, reducing the evidential strength of the conclusions.
Overall, the work is better characterized as a technical report rather than a scientific paper, as critical details are unclear and unsupported claims abound, making replication highly challenging.

**Strengths:**

Next to the actual weights (if made publicly available, not mentioned in the paper), the main contributions of this paper consist of the vaguely described data-set creation and agentic research simulation environment. The paper itself is clearly written and figures are neat and polished.
If correct and not biased, the results are very impressive. While this experimental report has nothing to do at a **scientific conference**, the effort of the authors to improve open source SOTA is of great value for any non-scientific user.

**Weaknesses:**

The Core Issue: The paper presents itself as an engineering or experimental report (i.e., "we built a system that achieves X performance") rather than a scientific paper that answers a fundamental research question. It lacks a clearly defined scientific problem (e.g., "To what degree does data structure, versus data scale, impact an agent's reasoning?"). This focus on the final artifact, rather than the underlying principles, makes it difficult for the scientific community to extract deep understanding, generalizable intuitions, or understand why this method is better than baselines. To address this main drawback, I advise the authors to:
* Reframe the Contribution: The paper should be reframed around a central scientific question, not just a final performance score.

* Provide Rigorous Ablation Studies: The paper proposes several new components at once (Agentic CPT, graph-based synthetic data, scalable RL). It is impossible to know which part contributed most to the performance. The paper must include ablations to isolate the impact of each component on the improvement.

Another core weakness of this paper is the vague descriptions of its key contributions. Overall, the clarity could be improved tremendously by: elaborating more detailed how the knowledge graph is built, how the data is sampled from the graph, how exactly the simulated environment works, how the reward is computed, etc.

Additionally, there are no attempts made at making the method reproducible.

Lastly, throughout the text is a number of claims that were not shown empirically, e.g
* Page 2, uncertainty in QA: “We also expand the diversity of our QA generation by incorporating a wider variety of uncertainty definitions beyond obfuscation, directly targeting the need for more sophisticated reasoning”
Where are those uncertainty definitions?
* Page 2, considering the simulated environment: “Through meticulous design, it achieves high fidelity, ensuring that the agent’s interaction dynamics, state transitions, and reward mechanisms closely mirror those of a real-world setting”
Never shown.
* Page 4, about the graph subsampling: “Ultimately, this strategy enables us to efficiently gather a sufficient quantity of non-isomorphic connected subgraphs that collectively represent the full spectrum of structural complexities, without the prohibitive cost of a brute-force search.”
Never shown, (and "non-isomorphic nodes" is never defined).
* Page 4, QA generation: “To this end, we introduce a wider array of defined uncertainties, aiming to elicit a more diverse and comprehensive suite of advanced reasoning abilities from the model”
It’s never tested if the reasoning abilities are actually more diverse or comprehensive
* Page 5, Data Curation: “ The quality of the training data directly establishes the upper bound for the model’s ability to generalize to out-of-distribution scenarios through self-exploration.”
Has this been shown somewhere?

Finally, while the figures are neat and the text clear. I'd advise the two following points to improve the clarity, but also improve the scientific accuracy of the paper:
Clearly include:
I. A list of contributions at the end of the introduction (often denoted with i., ii., ...) that stands out visually.
II. A list of scientific research questions at the beginning of the experimental evaluation section (often denoted RQ1, RQ2, ... etc), that are each answered in different paragraphs. For example
* (RQ1) Can our training pipeline help open source agents outperform existing research agents?
* (RQ2) Can the graph creations help external viewers to verify the integrity of the provided results?
* (RQ3) How important is component A ? (conducting an ablation study)
* ...

The figures and tables can thus help the readers understanding the addressed problem, the method, and the experimental evaluation during their first pass over the paper. They should thus be structuring the paper like, e.g.:
1. Figure 1 (describing the problem, if the problem is not obvious),
2. Figure 2 (describing the method),
3. Figures and Tables describing each important results (that answers precise scientific questions mentioned above).

Further, each caption should make its figure a standalone component of the paper. Thus, each figure should be built in the following way:

The first sentence should highlight the main message of the Figure/Table (e.g. "Our method outperforms the existing SOTA methods on the studied problem.")

The next sentences then explain what is depicted in the Table/Figure. E.g. Mean test accuracy, on 5 seeded trainings, with std.

Finally, details and references to e.g. appendix can be provided if necessary. E.g. Our method outperforms baseline 1 in 3 out of 4 tasks, ... etc.

**Questions:**

* Will you publish your code for the complete pipeline, incl. the building of the knowledge graph, sampling from it, the simulated training environment, the reward model, the tool-calling API, the training loop itself?
* How do you ensure that the simulated environment closely resembles the real world?
* Will you make the SailorFog-QA-2 dataset publicly available?

---

> ### Author Response · Authors · 2025-11-20
> **Response to Reviewer uNmb (1/2)**
>
> We appreciate the reviewer's insightful feedback and acknowledge the importance of distinguishing between an engineering report and a scientific paper.
>
> **Response to "The Core Issue: Engineering vs. Science"**
>
> > *The reviewer states: "The paper presents itself as an engineering or experimental report... rather than a scientific paper."*
>
> We believe that in the rapidly evolving field of LLMs, the line between engineering and science is inherently blurred. A state-of-the-art system that demonstrates exceptional performance is not merely an engineering feat but often represents a significant scientific discovery in itself. Our work contributes to the scientific understanding of LLMs in the following ways, which we believe are valuable to the research community:
>
> *   **Generalization of Reasoning Patterns:** Our findings reveal a crucial insight into the generalization capabilities of LLMs. By training our model exclusively on exceptionally complex reasoning data—such as the BrowseComp-style dataset, which involves information retrieval problems rarely encountered by humans—we achieved near-perfect accuracy on simpler knowledge-based QA tasks like Simple-QA. This suggests that the model learns underlying reasoning patterns that are transferable across tasks of varying complexity. This finding challenges the community to focus more on abstract reasoning patterns rather than the superficial characteristics of specific questions.
> *   **Data Quality & Environmental Stability as Dominant Factors:** Our research underscores the critical role of data quality and the stability of the RL environment. Throughout our experiments, we found that algorithmic innovations had a less significant impact on final performance compared to improvements in data quality and dynamic data adjustments. This provides strong evidence that for the current generation of LLMs, progress is not solely dictated by novel algorithms but is substantially influenced by data and infrastructure.
>     *   Consider PPO as an analogous example: since its introduction in 2017, numerous efforts have attempted to improve upon it, yet it remains the standard algorithm in RL. We wish to engage the research community on this critical issue: given the complexity of modern LLM training, evaluating algorithms without a baseline of high-quality data and robust systems can often lead to invalid conclusions.
> *   **Simplicity & "The Bitter Lesson":** We intentionally avoid complex context management (e.g., memory banks) and opt for the straightforward ReAct framework, yet we achieve performance fully comparable to methods that utilize extensive context engineering. Since context management often involves lossy compression, it is scientifically valuable to question whether the information loss outweighs the computational cost of long contexts. Guided by "The Bitter Lesson," we argue that frameworks requiring rigid, human-engineered structures may become obsolete as models' intrinsic capabilities scale.
>
> **Response to "Provide Rigorous Ablation Studies"**
>
> > *The reviewer mentions: "The paper proposes several new components at once (Agentic CPT...)"*
>
> We respectfully clarify a potential misunderstanding: the review mentions Continual Pre-Training (CPT), whereas our work is a complete **post-training pipeline** (SFT + RL), and we do not employ CPT in this work.
>
> Regarding the ablation of the RL component, this can be assessed based on the clear improvements contributed by SFT and RL respectively: after SFT, the score on BrowseComp is 24.4, and after RL it rises to 35.3. This demonstrates the distinct contribution of each stage. Because the overall training cost for end-to-end runs of a 30B MoE model is extremely high, conducting a fully systematic ablation study for every sub-component is computationally prohibitive. Our primary goal is to present a feasible, unified path toward building a search-reasoning model that surpasses human-level performance.

---

> > ### Author Response · Authors · 2025-11-20
> > **Response to Reviewer uNmb (2/2)**
> >
> > **Response to "Claims not shown empirically"**
> >
> > We appreciate the reviewer highlighting specific areas where definitions or evidence appeared lacking. We clarify these points below and will include these details in the revision:
> >
> > *   **Uncertainty Definitions:** In addition to obfuscation, we introduce specific new ways of injecting uncertainty, including: (1) *Noise with incorrect information* (distractors), (2) *Semantic ambiguity* (under-specified entities), and (3) *Reasoning with missing information* (derived from removing graph cut vertices and edges).
> > *   **The Simulated Environment:** We test our adapted GRPO algorithm in this environment and find that the resulting reward curve closely matches the one observed in the real environment. This Wiki-based simulation functions analogously to a "wind tunnel laboratory," enabling fast algorithm iteration and greatly improving development efficiency.
> > *   **Non-isomorphic Nodes:** This refers to nodes that occupy structurally different positions in a graph. For example, in a three-edge cycle, every node has the same structural role; in a three-node path, there are two distinct roles (the central node vs. the peripheral nodes). Analyzing this allows us to sample diverse reasoning paths.
> > *   **QA Generation Diversity:** We find that introducing these new uncertainties enables the model to exhibit a more diverse range of reasoning patterns. For instance, the model gains stronger *error-correction abilities*: it can initially judge a question incorrectly, realize the error, and pivot to a new line of reasoning. We also observe that while BrowseComp-style questions are hard to answer, they are easy to verify. Previously, the model often failed to verify the correct answer even when provided with it; after injecting specific noise-like data, this verification capability improved significantly. We will include an explanation of this phenomenon in the paper.
> > *   **Data Curation:** This refers to the dynamic replacement of training data during the RL process based on metrics such as reward and entropy. While specific implementation thresholds cannot be disclosed due to confidentiality, the methodology significantly contributes to preventing overfitting to low-quality trajectories.
> >
> > **Response to Reproducibility Questions**
> >
> > > *Will you publish your code... and make the SailorFog-QA-V2 dataset publicly available?*
> >
> > Yes, we will open-source the complete SailorFog-QA-V2 dataset and the associated data construction pipeline upon publication; this is currently going through our internal approval process. We do not plan to open-source the simulation environment at this time.
> >
> > **Response to Simulator Fidelity**
> >
> > > *How do you ensure that the simulated environment closely resembles the real world?*
> >
> > Our current draft indeed under-explains this point. In our internal experiments, we evaluated realism based on the **alignment of training dynamics** rather than absolute score matching.
> >
> > The simulated environment serves as an economical solution for algorithmic validation. While strict content consistency between the virtual and real environments is impossible, our experiments demonstrate a high degree of alignment in training dynamics. Specifically, we trained the agent in both the Wikipedia-based simulator and the real environment using the same algorithm. We tracked key metrics (Average Reward and Pass@1 Accuracy) and observed that the trends are highly correlated: algorithmic changes that improve performance in the simulator almost always yield similar improvements in the real environment. We will explicitly acknowledge the limitation regarding large-scale sim-to-real benchmarks in the revised manuscript.

---

> > ### Comment · Reviewer_uNmb · 2025-11-28
> > **Engineering is not science, but yes they are not completely distinct fields**
> >
> > Thank you for your comments. After reading your answer, I would like to follow up the discussion on engineering vs science.
> >
> > The current trend of adapting LLM or expanding their abilities makes that ICLR, among other *scientific* conferences, gets more and more engineering reports published. The distinction I am making between these science and engineering is not clear in my comment, so I will try to provide precisions.
> >
> > I am both part of academia and industry (most of my time spent in industry), and I do greatly value the benefits of projects like WebSailor-V2. These are amazing contributions to the advancement of applied AI (or here LLMs). If I were to come to a trade fair, or an industry meetup, I would love seeing WebSailor there and would probably spend lots of time discussing with its authors. Similar, if I am looking for the best FOSS product, I would be glad and thankful to find WebSailor on the market.
> >
> > However, when I go to a scientific conference, I expect to see and discuss scientific contributions. This means that:
> > * I am presented a clearly identified problem (not "we pushed SOTA", but existing systems fail on this specific problem (hence capping their performances on benchmark X).
> > * I am presented a method that clearly addresses the identified problem, and this method is clearly explained. Following the authors reasoning, I get insight of: "If I were to be facing this issue in another field, I can now better identify the underlying reason(s) and the method would obviously solve it".
> >
> > That means if I know is I should accept the paper ***before*** I see the experimental evaluations, that supports the -(clearly identified) authors claim. I do not remember the experimental evaluation's details, I remember the technique used to address the scientific problem.
> >
> > To take your PPO example. I remember that PPO solves the problem of large, destructive policy updates by employing a clipped surrogate objective function (clipping instead of using KL-divergence as TRPO does, as its simpler) to constrain the change between the new and old policies, ensuring stable and monotonic policy improvement. I have no clue anymore about the improvements scores achieved by PPO on the used benchmark (I don't even remember what benchmark they used). I just know that the simple clipping strategy experimentally works.
> >
> > Now, I am not saying that writing a scientific paper that explains exactly what WebSailor addresses and why it does it that way, is not doable, but this is, from what I can see, not the state of the submitted manuscript.
> >
> > You do not need many changes to make turn your provided amazing work into a scientific paper. You simply need to:
> > * clearly expose the problem(s)
> > * clearly list the claims -> your solutions address the problem(s).
> > * back up the claims with experimental support (where you can rightfully brag about SOTA, but you're likely to keep SOTA for a couple of months, at best).
> >
> > Having said all that. You answered my other concerns and most of the other reviewers one (which do not seem to care about the difference between research and engineering). I am thus willing to move my score, ideally to 5, but as not feasible probably 6 as the other reviewers value your work.
> >
> > You can still modify your paper (if you will), to better incorporate this clear problem isolation and better expose your future readers with to the scientific insight/claims that your experimental evaluation supports.

---

> > > ### Author Response · Authors · 2025-11-29
> > > **Response to Reviewer uNmb**
> > >
> > > We are genuinely grateful for your constructive follow-up. Your explanation of the distinction between an engineering report and a scientific paper was incredibly illuminating. **We are also greatly encouraged by your willingness to raise the score to 6 based on these revisions.** Below, we detail the specific revisions made to the **Introduction**, **Experiments**, and **Conclusion** sections in the current manuscript.
> > >
> > > ### 1. Clear Exposure of the Scientific Problem & Claims (Introduction)
> > >
> > > We have rewritten the contribution paragraph to define the underlying barriers (insufficient diversity in linear data & instability in RL) and frame our methods as specific scientific strategies to solve them.
> > >
> > > > "In this paper, we present WebSailor-V2, a comprehensive framework designed to overcome these barriers through topological data synthesis and environmental stabilization.
> > > >
> > > > **(1) Topological Strategy for Reasoning:** To resolve the insufficient diversity inherent in linear or tree-like training data, we introduce **SailorFog-QA-V2**, an enhanced dataset built upon SailorFog-QA (Li et al., 2025b). Unlike conventional “easy-to-hard” expansion, our method constructs densely interconnected knowledge graphs. By specifically sampling trajectories that require resolving cyclic dependencies and traversing critical “cut vertices”, we force the agent to learn abstract graph-search patterns. This mechanism ensures the model acquires robust reasoning capabilities that generalize beyond simple information retrieval.
> > > >
> > > > **(2) Stabilization Strategy for RL:** To mitigate the “signal-to-noise” ratio issue caused by real-world volatility, we propose a **Symbiotic Dual-Environment framework**. By utilizing a high-fidelity simulator as an algorithmic “Wind Tunnel”, we decouple policy learning from environmental stochasticity. This allows for high-frequency, stable policy iteration, ensuring that the agent establishes a robust policy before adapting to the noisy, managed real-world interface."
> > >
> > > ### 2. Structuring Evaluation around Research Questions
> > >
> > > We introduced explicit Research Questions (RQs) to guide the evaluation.
> > >
> > > > "**Research Questions** To verify our methodological claims, our evaluation is designed to answer two core scientific questions:
> > > >
> > > > *   **(RQ1) Topological Generalization:** Does training on data derived from dense, cyclic knowledge graphs (SailorFog-QA-V2) yield stronger reasoning capabilities compared to standard linear or tree-like data expansion used by baselines?
> > > > *   **(RQ2) Environmental Stability:** Does decoupling training into a dual-environment (Simulation-to-Real) framework effectively mitigate the instability of real-world RL and lead to better convergence?"
> > >
> > > ### 3. Explicitly Linking Evidence to Claims
> > >
> > > We updated the caption of the main result table to explicitly state it as evidence for RQ1.
> > >
> > > > "**Table 1: Graph-based training data enables superior reasoning generalization (Evidence for RQ1).** WebSailor-V2 outperforms baselines trained on linear/tree-like synthetic data by a significant margin on complex benchmarks like BrowseComp."
> > >
> > > We updated the analysis in Section 5.4 to explicitly link the training dynamics (Figure 4) to RQ2.
> > >
> > > > "As illustrated, **stabilized environments lead to consistent policy improvement (Evidence for RQ2).** The clear upward trend in reward (left) confirms that our dual-environment approach successfully mitigates the “destructive updates” often seen in noisy real-world RL."
> > >
> > > ### 4. Summarizing Scientific Insights (Conclusion)
> > >
> > > We rewrote the conclusion to emphasize the validation of our scientific hypotheses rather than just the engineering success.
> > >
> > > > "In this work, we present WebSailor-V2... Beyond achieving state-of-the-art performance with a 30B model, our work validates two critical scientific insights regarding agentic learning:
> > > >
> > > > **(1) Topological Generalization:** We demonstrate that the logical structure of training data is as critical as its scale. By shifting from standard linear data expansion to cyclic, graph-based synthesis (SailorFog-QA-V2), we show that forcing agents to navigate dense, interdependent relationships during training significantly enhances their ability to generalize to complex, multi-step reasoning tasks.
> > > >
> > > > **(2) Environmental Stability in RL:** We identify environmental stochasticity as a primary bottleneck for on-policy reinforcement learning. Our results confirm that a symbiotic dual-environment framework—decoupling algorithmic learning in a high-fidelity simulator from adaptation in the managed real world—is essential for mitigating destructive updates and ensuring stable convergence.
> > > >
> > > >Ultimately, WebSailor-V2 proves that ..."
> > >
> > > We believe these changes effectively transform the paper from an engineering report into a scientific study that isolates and validates the impact of **Data Topology** and **Environmental Stability**. We deeply appreciate your guidance in this process.

---

### Official Review · Reviewer_YWs5 · 2025-10-31

**Soundness:** 3
**Presentation:** 3
**Contribution:** 3
**Rating:** 8
**Confidence:** 3

**Summary:**

This paper presents WebSailor-V2, a complete post-training pipeline (SFT + RL) designed to bridge the performance gap between open-source and proprietary web agents.

**Strengths:**

1. Motivated Data Construction: The paper identifies a clear weakness in existing agent datasets (acyclic, tree-like structures and over-reliance on simple obfuscation). The proposed SailorFog-QA-V2, with its densely interconnected, cyclic graph and "wider array of uncertainties," is a strong and novel contribution that better reflects the complexity of real-world information.

2. The dual-environment RL approach is a very practical and well-executed solution to a major bottleneck in agent training. The high-fidelity simulator (using an offline Wikipedia dump) allows for rapid, low-cost algorithmic iteration, while the robust real-world environment, featuring a unified tool interface with fault tolerance and retry mechanisms, insulates the final training from the latency and failures of live APIs. This is a significant engineering contribution that enables stable RL.

3. The results are impressive and clearly demonstrate the efficacy of the proposed pipeline. The fact that the 30B WebSailor-V2 significantly outperforms all other open-source agents and even much larger models like the 671B DeepSeek-V3.1 validates the hypothesis that superior data and training strategies can overcome raw model scale.

**Weaknesses:**

Could the authors elaborate on the specific types of "uncertainties" (beyond obfuscation) that were introduced in the SailorFog-QA-V2 dataset? Providing a few examples would greatly clarify this contribution.

**Questions:**

See weakness.

---

> ### Author Response · Authors · 2025-11-20
> **Response to Reviewer YWs5**
>
> Thank you very much for your highly encouraging review. We are truly grateful that you found the motivation, design, and results of WebSailor-V2 convincing, and we especially appreciate your positive assessment of both the SailorFog-QA-V2 dataset and the dual-environment RL framework. Your summary captures exactly what we hoped to convey—namely, that carefully designed data and training strategies can substantially narrow the gap between open-source agents and large proprietary systems—and your endorsement is extremely motivating for us.
>
> Regarding your request for more detail on the "uncertainties" beyond obfuscation in SailorFog-QA-V2, we fully agree that concretizing these definitions and giving examples will make this contribution much clearer. In the revised version, we will expand Section 3.3 with a dedicated paragraph that enumerates uncertainty types. Below, we summarize the main categories and their intended effects on agent behavior:
>
> 1.  **Semantic obfuscation through contextual ambiguity:**
>     We deliberately formulate questions where key references are under-specified or only described indirectly—for example, referring to "the earlier of the two major conferences where this work was first presented" instead of naming the venue, or using a role description (e.g., "the mathematician who later generalized this result") rather than the explicit entity name. These cases require the agent to (i) reason to reconstruct the missing entity from the ambiguous context, and (ii) verify that it has resolved the ambiguity correctly before answering. This goes beyond obfuscation by making the mapping from description to entity itself non-trivial.
>
> 2.  **Noise with incorrect information:**
>     Beyond simply masking or paraphrasing key entities (standard obfuscation), we sometimes inject factually incorrect distractors into the subgraph or the retrieved evidence. For instance, when the gold answer involves the publication year of a specific paper, we may insert another year that corresponds to a related work by the same authors, or when the task concerns a particular scientist’s affiliation, we include a closely related institution that appears frequently in the same neighborhood of the graph. This forces the agent not only to "find something that looks right" but to cross-check multiple sources and actively discard inconsistent evidence, exercising contradiction detection and error correction rather than relying on shallow pattern matching.
>
> 3.  **Structural analysis of non-isomorphic subgraph structures:**
>     A key contribution is our systematic investigation of how non-isomorphic subgraph structures influence QA difficulty and reasoning patterns. Using a random-walk based approach with Weisfeiler-Leman verification, we extract diverse subgraphs and analyze their structural complexities. We introduce controlled uncertainties by strategically removing critical nodes/edges (e.g., cut vertices/bridges) from sampled subgraphs. This requires the agent to reconstruct multi-hop relations through alternative paths or knowledge graph redundancy, encouraging global, topology-aware reasoning rather than local adjacency dependence. These uncertainty types are distributed across orbit nodes (different structural roles) to ensure comprehensive reasoning coverage without overfitting to specific patterns.
>
> These uncertainty types are instantiated systematically over non-isomorphic structural roles in the subgraphs, so that the resulting QA set covers a broad range of reasoning configurations rather than overfitting to a single "style" of uncertainty. We will incorporate these clarifications and illustrative examples into the camera-ready version by explicitly listing and defining the uncertainty categories in Section 3.3. Once again, we sincerely thank you for your generous evaluation and for pointing out this opportunity to better highlight a key component of SailorFog-QA-V2; your feedback directly improves the clarity and impact of our work.

---

### Official Review · Reviewer_Usyc · 2025-11-01

**Soundness:** 3
**Presentation:** 3
**Contribution:** 3
**Rating:** 4
**Confidence:** 4

**Summary:**

The paper introduces Sailor-Fog-QA-V2, a refreshed data generator that builds denser, cyclic knowledge graphs and samples non-isomorphic subgraphs via random-walks, plus a dual-environment RL setup (offline Wikipedia simulator + robust real-web tools stack). With SFT on synthetic trajectories from V2 and a GRPO-style RL phase, a Qwen3-30B-A3B agent delivers SOTA among open-source and competitive vs proprietary systems on BrowseComp-EN/ZH, xbench-DeepSearch, GAIA, HLE, and DeepResearch Bench (e.g., 35.3 on BrowseComp-EN, 44.1 on BrowseComp-ZH, 30.6 on HLE).

**Strengths:**

* Sailor-Fog-QA-V2 deliberately creates cyclic, densely connected graphs (vs tree-like) and preserves procedural metadata (queries, URLs, stats) that supports richer QA targeting across orbit nodes. Subgraphs are random-walk sampled and WL-checked for non-isomorphism.
* A Wikipedia-based simulator enables fast, low-cost iteration, whereas the real environment adds a unified tool interface with QPS controls, caching, retry/timeout, and fallback switching, which directly addresses brittleness in web-tool rollouts.
* The 30B-A3B MoE agent surpasses prior open-source, narrows the gap with leading proprietary systems, and even beats DeepSeek-V3.1 (671B) on HLE, showing that data and stability matter as much as model scale.

**Weaknesses:**

* The paper states it goes “beyond obfuscation” but never enumerates the additional uncertainty types or how they’re realized during QA generation (only a general description of obfuscation appears). This makes it hard to attribute gains to the proposed “wider variety.”
* The dual-environment design is described, but how rollouts are scheduled/allocated between simulator and real web (by phase, by task type, by curriculum, or adaptively from reward signals) is not specified. Figure 2 shows the loop but not the selection strategy.
* Training uses Sailor-Fog-QA (v1), V2, and IterBench, yet there’s no quantitative ablation showing the marginal benefit of each component on the benchmarks.

**Questions:**

1. Please list and define the uncertainty categories added beyond obfuscation, with 1–2 concrete generation templates per type. Where in the pipeline are they injected?
2. Are RL rollouts taken in both simulator and real web? If yes, what mix over training (e.g., % rollouts or steps per phase), and do you adapt the mix based on validation reward or environment-specific success rates?
3.  You drop some negative trajectories to avoid format collapse. Which categories are excluded beyond “no final answer due to length,” and how sensitive is performance to this filter?

---

> ### Author Response · Authors · 2025-11-20
> **Response to Reviewer Usyc (1/2)**
>
> We thank Reviewer Usyc for the careful reading of our paper and for the constructive, technically detailed feedback. Below we address each concern point by point and will incorporate the corresponding clarifications into the revised manuscript.
>
> **1. "Beyond obfuscation": uncertainty categories and templates**
>
> We agree that the current draft does not sufficiently spell out the additional uncertainty types beyond obfuscation. In the revision, we will explicitly list and formalize the categories used in SailorFog-QA-V2. Below we summarize the main categories and provide examples:
>
> 1.  **Semantic obfuscation through contextual ambiguity:**
>     We deliberately formulate questions where key references are under-specified or only described indirectly—for example, referring to "the earlier of the two major conferences where this work was first presented" instead of naming the venue, or using a role description (e.g., "the mathematician who later generalized this result") rather than the explicit entity name. These cases require the agent to (i) reason to reconstruct the missing entity from the ambiguous context, and (ii) verify that it has resolved the ambiguity correctly before answering. This goes beyond obfuscation by making the mapping from description to entity itself non-trivial.
>
> 2.  **Noise with incorrect information:**
>     Beyond simply masking or paraphrasing key entities (standard obfuscation), we sometimes inject factually incorrect distractors into the subgraph or the retrieved evidence. For instance, when the gold answer involves the publication year of a specific paper, we may insert another year that corresponds to a related work by the same authors, or when the task concerns a particular scientist’s affiliation, we include a closely related institution that appears frequently in the same neighborhood of the graph. This forces the agent not only to "find something that looks right" but to cross-check multiple sources and actively discard inconsistent evidence, exercising contradiction detection and error correction rather than relying on shallow pattern matching.
>
> 3.  **Structural analysis of non-isomorphic subgraph structures:**
>     A key contribution is our systematic investigation of how non-isomorphic subgraph structures influence QA difficulty and reasoning patterns. Using a random-walk based approach with Weisfeiler-Leman verification, we extract diverse subgraphs and analyze their structural complexities. We introduce controlled uncertainties by strategically removing critical nodes/edges (e.g., cut vertices/bridges) from sampled subgraphs. This requires the agent to reconstruct multi-hop relations through alternative paths or knowledge graph redundancy, encouraging global, topology-aware reasoning rather than local adjacency dependence.
>
> **2. Dual-environment RL: rollout allocation and adaptation**
>
> We appreciate the request for more concrete details on how the simulated and real environments are used. Our intention is to use the simulator primarily as a "wind-tunnel" for algorithm and reward design, and the real environment for the final large-scale training run.
>
> **Are RL rollouts taken in both environments?**
> Yes, but in different capacities:
> *   In the simulated Wikipedia-based environment, we perform numerous shorter RL runs to tune GRPO-style hyperparameters (e.g., clipping ranges, learning rate schedule), test different reward shaping variants and trajectory filtering policies, and validate that the training dynamics (reward and pass@1) are stable.
> *   For the final WebSailor-V2-30B-A3B agent reported in Table 1, the production RL run is carried out in the real environment only, starting from the SFT checkpoint.
>
> **Mix over training and adaptation.**
> In the current work, we do not interleave simulator and real-world rollouts within a single training run, nor do we adapt the mix online based on validation rewards. Instead, we follow a staged procedure:
> 1.  **Algorithm design in simulation:** We run multiple RL experiments entirely in the simulator until we obtain a configuration whose reward and pass@1 curves are stable and whose trend matches that observed in smaller-scale real-environment tests.
> 2.  **Single large-scale RL run in the real environment:** Once the configuration is fixed, we run one large-scale training job entirely in the real web environment. Given the engineering and cost constraints (high QPS, fault tolerance, and tool heterogeneity), performing multiple full real-environment runs for different mixes of sim/real rollouts is prohibitive for this 30B MoE model.
>
> We will clarify this staged strategy in the paper.

---

> > ### Author Response · Authors · 2025-11-20
> > **Response to Reviewer Usyc (2/2)**
> >
> > **3. On ablations for SailorFog-QA (v1), SailorFog-QA-V2, and IterBench**
> >
> > We agree that, conceptually, it would be very informative to isolate the marginal effect of each data component. However, running a full set of data-source ablations at the same 30B MoE scale would require multiple end-to-end SFT+RL runs, each involving many millions of tool calls and substantial wall-clock time. This is currently beyond our computational budget.
> >
> > That said, we will improve the paper by providing a qualitative breakdown in the experiments section discussing the role of each dataset:
> > *   **SailorFog-QA (v1):** Provides broad coverage of web-style reasoning, serving as the base scaffold.
> > *   **SailorFog-QA-V2:** Introduces denser, cyclic graph structures and the new uncertainty patterns described above, which we find crucial for challenging BrowseComp-style tasks.
> > *   **IterBench:** Specifically strengthens mathematical and academic reasoning, primarily reflected in HLE and DeepResearch Bench performance.
> >
> > We will also make the computational constraints explicit in the limitations section, so the absence of full-scale ablations is clearly framed as a practical trade-off rather than an oversight.
> >
> > **4. Negative trajectory filtering: categories and sensitivity**
> >
> > Thank you for your question regarding our negative trajectory filtering strategy. Regarding your question on which categories are excluded, we only excluded one specific category of negative samples: those that "do not yield a final answer due to length limitations." For these cases, we mask the loss of the entire trajectory to prevent gradient backpropagation during training.
> >
> > For trajectories with obvious format errors, we handle them differently by assigning negative rewards to penalize such behaviors rather than excluding them entirely. This approach allows the model to learn from these mistakes while still discouraging poor formatting. The key distinction is that for excessively long trajectories that do not contain format errors but exceed length limits without producing a final answer, defining appropriate rewards is challenging; therefore, we conservatively mask their losses as a precautionary measure.
> >
> > Regarding sensitivity, our experiments clearly demonstrate the critical importance of this filter. Without this mechanism, models typically experience format collapse around 100 training steps. In contrast, implementing this simple yet effective filter successfully prevents format collapse entirely, maintaining stable training dynamics throughout the entire training process.
> >
> > We hope these clarifications address your concerns and demonstrate that the reported gains are tightly connected to the specific design choices in SailorFog-QA-V2 and our dual-environment RL framework.

---

### Official Review · Reviewer_rnrd · 2025-11-01

**Soundness:** 3
**Presentation:** 3
**Contribution:** 3
**Rating:** 6
**Confidence:** 3

**Summary:**

The paper presents WebSailor-V2, an open-source web agent trained through a multi-stage post-training pipeline. The approach combines a dataset generated from a knowledge-graph-based process intended to cover diverse forms of uncertainty, supervised fine-tuning on synthetic demonstrations, and reinforcement learning within a dual-environment setup, consisting of a Wikipedia-based simulated environment and a managed real-world environment with tool-execution interfaces. The model is based on Qwen3 and uses 128k-token contexts. The RL stage employs a GRPO-style algorithm with token-level updates, leave-one-out advantages, on-policy sampling, and filtering of negative trajectories. The paper reports that this combination improves performance relative to SFT-only baselines and narrows the gap between open-source and proprietary web agents.

**Strengths:**

- Clear description of a complete training pipeline.
- Reported benchmark gains on BrowseComp and HLE relative to earlier open models.
- RL procedure and stability considerations are explicitly stated.
- Diagnostic plots provide visibility into training behavior.

**Weaknesses:**

- The paper provides no quantitative validation that the simulated environment approximates real-world web interactions.
-  The paper does not report SailorFog statistics such as number of QA pairs, node/edge counts, or category distributions.
- The effect of individual components (dataset, SFT, RL) and RL design choices is not shown beyond one aggregated comparison.
- Reported comparisons to proprietary systems use previously published numbers without matching evaluation conditions (context length, sampling, tool limits).
- Accuracy is measured with an LLM-based judge and the paper does not provide calibration or human-agreement checks.
- It is not explicitly stated whether code, data, and simulator will be released concurrently with publication.

**Questions:**

- What are the quantitative properties of SailorFog (size, graph statistics, uncertainty-type counts)?
- How is simulator realism assessed? Are there any metrics comparing simulated and real tool outputs?
- Which parts of the performance gain come from SFT data versus RL fine-tuning?
- Were all benchmark runs conducted under identical context lengths and tool budgets?
- How consistent is the LLM-as-judge evaluation, has human validation been performed?
- Will the dataset, simulator, and training code be publicly released?

---

> ### Author Response · Authors · 2025-11-20
> **Response to Reviewer rnrd (1/2)**
>
> We thank the reviewer for the thoughtful and constructive feedback, and we are glad that you found the pipeline description, benchmark gains, and diagnostic plots useful. Below we respond to each concern in detail and will incorporate the corresponding clarifications, statistics, and discussions into the revised version of the paper.
>
> **1. On the realism of the simulated environment**
>
> The reviewer correctly notes that our current draft under-explains the quantitative validation of the simulator. In our internal experiments, we evaluated realism along the axis of training-dynamics alignment rather than absolute score matching.
>
> The simulated environment we propose serves as a more economical and efficient solution for algorithmic and data validation. While strict consistency between the virtual and real environments cannot be guaranteed due to inherent tool differences and the resulting trajectory variations, our experiments clearly demonstrate a high degree of alignment in training dynamics between the two environments. This alignment suggests that data and algorithms extensively validated in the virtual environment are more likely to exhibit favorable behaviors in the real environment.
>
> Specifically, we trained the agent in both the Wikipedia-based simulated environment and the real environment using the same GRPO-style algorithm and comparable hyperparameters. For a set of shared evaluation suites (e.g., held-out SailorFog-style tasks and a BrowseComp-style subset), we tracked several key metrics, including average reward and Pass@1 accuracy. We observed that the trends of these metrics are highly correlated between the simulator and the real environment: when a particular change in policy or reward shaping improves reward and pass@1 performance in simulation, we almost always observe changes in the same direction in the real environment, although the absolute values may differ.
>
> Due to computational and cost constraints associated with real-environment rollouts, we are unable to expand this into a full large-scale sim-to-real study with multiple seeds and comprehensive benchmarks. We will explicitly acknowledge this limitation in the revised manuscript.
>
> **2. Quantitative properties of SailorFog-QA-V2**
>
> We agree that this information is important for interpretability and reproducibility. In the revised paper, we will add a dedicated subsection summarizing the following key statistics:
> *   **Graph statistics (per connected component used for QA generation):**
>     *   Average number of nodes: $\approx$ 30.
>     *   Average degree: $\approx$ 2.5.
>     *   The random-walk sampling with Weisfeiler-Leman based non-isomorphism checks yields a diverse set of subgraphs that cover a broad range of topologies (chains, trees with branches, cycles, and small dense subgraphs).
> *   **Dataset scale:**
>     *   Number of QA pairs used in SFT + RL: 30k+ (SFT and RL sampling draw from the same SailorFog-QA-2 generation pipeline but with different filtering).
> *   **Uncertainty types:**
>     *   Semantic obfuscation through contextual ambiguity.
>     *   Noise with incorrect information.
>     *   Structural complexities derived from different non-isomorphic subgraph structures.
>
> **3. Effect of individual components and RL design choices**
>
> We appreciate the reviewer's concern regarding component isolation. In a complex, multi-stage, resource-intensive pipeline like ours, a fully exhaustive ablation (e.g., varying SFT steps $\times$ RL variants $\times$ data mixtures across many seeds) is unfortunately beyond our computational budget. Different choices are also coupled; for instance, training SFT "more thoroughly" tends to reduce the headroom that RL can exploit, meaning the marginal benefit of RL is dependent on earlier stages.
>
> That said, we can be clearer about the available evidence regarding the Data vs. RL contribution. The SFT-only model already achieves 24.4 on BrowseComp-EN and 23.9 on HLE. After RL, scores increase to 35.3 and 30.6, respectively. This demonstrates that both data (SFT) and RL are crucial: high-quality SailorFog-QA-V2 provides a strong policy initialization, while RL adds a substantial 7–11 point improvement on the more challenging benchmarks. We will explicitly discuss these contributions and clarify that a fully systematic study of all components is left for future work due to the high end-to-end training cost.

---

> > ### Author Response · Authors · 2025-11-20
> > **Response to Reviewer rnrd (2/2)**
> >
> > **4. Evaluation conditions for proprietary comparisons**
> >
> > Thank you for pointing out that our evaluation conditions for proprietary systems were not sufficiently documented. Our comparisons adhere to the following principles:
> > *   **Deployed Models:** For models that we manually deployed or accessed via API, we maintained consistent evaluation conditions by using a 128k context token limit, a maximum of 100 tool calls, and the standardized toolset defined in Section 2 of our paper. This uniform setup was applied particularly when the original model did not provide open-source configurations or specified tool limitations.
> > *   **Published Results:** For benchmarks where we relied on published results (e.g., certain BrowseComp scores for proprietary systems), we strictly followed the evaluation protocols specified in the corresponding benchmark papers or official system cards. To ensure transparency, we clearly marked these externally sourced numbers in Table 1 using distinct symbols (e.g., $\dagger$/$\ddagger$).
> > We will provide a more detailed description of these evaluation settings in the revised draft.
> >
> > **5. LLM-as-judge calibration and human validation**
> >
> > We appreciate your concern about calibration and human-agreement checks. First, we utilized the official evaluation prompts and models provided by each benchmark dataset. This approach ensures better comparability with previously published scores, as the vast majority of publicly reported results follow the official evaluation protocols specified by the benchmark creators.
> >
> > Second, to validate the reliability of our LLM judge, we conducted comprehensive manual validation across multiple random subsets of the BrowseComp-EN, BrowseComp-ZH, and HLE datasets. Our human evaluators carefully reviewed these samples, and we found that the final scores showed strong agreement with the LLM judge's assessments. This consistency gives us confidence in the reliability of our automated evaluation approach. In the revised version, we will add a short "Judge calibration" paragraph in the evaluation section summarizing the Human–LLM agreement percentages and clarifying that the LLM-judge configuration is fixed across all systems.
> >
> > **6. Release of code, data, and simulator**
> >
> > This is an omission in the current draft; thank you for prompting us to clarify. We will release the SailorFog-QA-V2 dataset upon acceptance, subject to our company’s standard internal review process, which is already underway. We will add an explicit "Open-sourcing plan" paragraph in the conclusion/experiments section summarizing these commitments.

---

### Author Response · Authors · 2025-11-20
**General Response to All Reviewers**

We thank all reviewers for their thoughtful feedback and for recognizing the strong performance of WebSailor-V2, particularly its ability to outperform significantly larger models on complex benchmarks like HLE and BrowseComp. **We have revised the manuscript based on the reviewers' comments, with all changes highlighted in blue.**

We have identified several common questions regarding (1) the specific definitions of "uncertainties", (2) the validation of the simulated environment, (3) the scientific nature of our contributions, and (4) reproducibility and open-source plans. We address these centrally here.

**1. Clarifying "Uncertainties Beyond Obfuscation"**

Reviewers asked for concrete definitions of the uncertainties introduced in SailorFog-QA-V2. We go beyond simple entity masking (obfuscation) to include the following mechanisms:
*   **Semantic Ambiguity:** We intentionally under-specify key entities (e.g., describing a person as "the mathematician who generalized this result" instead of providing their name) or dates. This forces the agent to reason over the graph structure to disambiguate entities before retrieving facts, rather than relying on keyword matching.
*   **Distractor Noise:** We inject plausible but factually incorrect distractors (e.g., a publication year of a related paper by the same author) into the retrieved context. This requires the agent to perform cross-verification and contradiction detection.
*   **Structural Constraints (Topology-aware Reasoning):** By analyzing non-isomorphic nodes (nodes with distinct structural roles, such as a bridge node versus a leaf node) via Weisfeiler-Leman hashing, we identify critical paths. We then generate questions that require traversing specific "cut" edges—paths that, if missed, render the answer unreachable. This forces global graph exploration rather than local greedy search.

We expand Section 3.3 to explicitly define and categorize these mechanisms.

**2. Validation of Simulated Environment**

Reviewers queried how we ensure the Wikipedia-based simulator resembles the real world.
*   **Goal:** Our simulator acts as a "wind tunnel" for algorithmic stability and reward shaping, rather than a direct replica of the live web's content.
*   **Alignment of Dynamics:** Our internal validation shows a high correlation in training dynamics (specifically Reward and Pass@1 improvement trends) between the simulator and the real environment. Algorithmic improvements that stabilize learning in the simulator consistently translate to stability in the real world.
*   **Efficiency:** This setup allows us to debug the RL loop (e.g., preventing format collapse, tuning GRPO clipping) at a low cost before committing to the expensive real-world training run. We explicitly discuss this alignment validation and its role in our staged training strategy in Section 4.2.

**3. Scientific Contribution vs. Engineering Report**

We respectfully argue that WebSailor-V2 contributes significant scientific insights beyond being a "system report":
*   **Data-Driven Generalization:** We demonstrate that training on highly complex, structurally dense synthetic data (SailorFog-QA-V2) enables generalization to simple, factual QA tasks. This provides evidence that LLMs can learn abstract reasoning patterns (such as error correction and pathfinding) that transfer across task complexities.
*   **System Stability as a Prerequisite for RL:** Our findings challenge the community's focus on novel loss functions. We show that standard GRPO fails without high-quality data initialization (SFT) and a stable environment (robust retry/fallback mechanisms). The scientific takeaway is that environmental stability and data density are the dominant factors in determining agentic RL success.
*   **Simplicity & The Bitter Lesson:** We purposely avoid complex context management (such as memory banks or compression), opting for vanilla ReAct. Achieving SOTA with this simple architecture suggests that as models scale, rigid human-engineered structures may be unnecessary or even detrimental.

**4. Ablations & Open Source**

*   **Ablation Constraints:** Due to the prohibitive cost of training 30B MoE models end-to-end, we focus on the clear delta between our SFT baseline (24.4 on BrowseComp) and the RL-tuned model (35.3), proving the efficacy of the pipeline.
*   **Release Plan:** We will release SailorFog-QA-V2 upon acceptance, subject to our company’s standard internal review process, which is already underway.

---

### Author Response · Authors · 2025-12-02
**Summary of Author Response and Reviewer Consensus**

We sincerely thank the reviewers for their constructive feedback and the Area Chair for coordinating this process. We are highly encouraged by the active engagement during the rebuttal phase, which led to a **consensus on the scientific value of our methodological contributions** and a significant improvement in assessment, most notably from **Reviewer uNmb (Score: 2 $\rightarrow$ 6)**.

Below is a summary of the major concerns raised and how our responses have strengthened the scientific rigor of the manuscript.

### **1. Addressed Key Concerns & Methodological Clarifications**

We have incorporated extensive revisions to ensure the paper reads not as an engineering report, but as a scientific study isolating the variables of **Data Topology** and **Environmental Stability**.

| Dimension | Reviewer(s) | Key Concern & Our Resolution |
| :--- | :--- | :--- |
| **Scientific Positioning & Framing** | **uNmb** (Score: $\uparrow$ 6) | **Concern:** The initial draft lacked a central scientific problem, appearing more as an engineering report. **Resolution:** We fundamentally restructured the paper to investigate two specific Research Questions (RQs): **(1) Topological Generalization:** Does dense, cyclic graph data yield stronger reasoning than linear expansion? **(2) Environmental Stability:** Does decoupling RL from real-world stochasticity mitigate destructive updates? Reviewer uNmb acknowledged this shift, raising their score to 6 as we successfully isolated these scientific insights from the engineering artifacts. |
| **Data Methodology: Beyond Obfuscation** | **Usyc, YWs5** | **Concern:** Requested concrete definitions of the "wider variety of uncertainties" introduced in SailorFog-QA-V2. **Resolution:** We formalized three specific uncertainty types injected to force sophisticated reasoning: **(1) Semantic Ambiguity:** Under-specified entities (e.g., "the mathematician who generalized this result") forcing contextual disambiguation rather than keyword matching. **(2) Distractor Noise:** Plausible but factually incorrect distractors to enforce active cross-verification. **(3) Structural Constraints:** Strategic removal of "cut vertices" in the graph, compelling the agent to perform global graph traversal to reconstruct multi-hop relations. |
| **Simulator Validity: The "Wind Tunnel"** | **rnrd, uNmb** | **Concern:** Lack of quantitative validation for the Wikipedia-based simulator's realism. **Resolution:** We clarified that the simulator functions as an algorithmic **"Wind Tunnel"** rather than a content replica. We provided evidence that **training dynamics (Reward and Pass@1 trends)** in the simulator are highly correlated with the real environment. This validates our symbiotic framework: algorithmic stability is solved in the simulator, ensuring the real-world run focuses purely on domain adaptation without destructive policy collapse. |
| **Component Attribution & Cost** | **Usyc** | **Concern:** Request for fine-grained ablations of every data component. **Resolution:** We clarified that while end-to-end ablations for 30B MoE models are computationally prohibitive (a constraint accepted by reviewers), the attribution is clear: **SFT on SailorFog-V2** establishes the fundamental reasoning scaffold (Score: 24.4), while the **Stabilized RL** acts as the critical adaptation phase (+11 points to 35.3). We also detailed our "negative trajectory masking" strategy, which was proven essential to prevent format collapse during RL. |

### **2. Performance as Scientific Validation**
Our empirical results serve as validation for our core scientific hypothesis: **High-density data topology and a stabilized training environment are the primary drivers for closing the gap between open-source and proprietary models.**
*   **Topological Generalization:** By training on cyclic, densely connected graphs, our **30B WebSailor-V2** achieves **35.3 on BrowseComp-EN** and **30.6 on Humanity’s Last Exam (HLE)**.
*   **Efficiency vs. Scale:** Crucially, this performance surpasses significantly larger models, including the **671B DeepSeek-V3.1** on HLE. This confirms that exposing the model to complex topological dependencies during training effectively substitutes for massive parameter scale when solving multi-step reasoning tasks.

### **3. Reproducibility and Open Source**
To facilitate future research and verify our findings:
*   **Dataset:** We are committed to open-sourcing the **SailorFog-QA-V2 dataset** and its generation pipeline upon acceptance.

We believe the revised manuscript now stands as a rigorous scientific contribution that not only delivers a strong model but also provides a reproducible methodology for democratizing deep research capabilities. We hope this summary assists the AC in their final decision.

Sincerely,

Authors of submission 17714

---

### Meta-Review · Program_Chairs · 2026-01-10

**Summary:**

This submission proposes a new open-source web agent based on the Qwen3-30B model. The authors utilize a post-training pipeline featuring SailorFog-QA-V2 (a synthetic dataset built on dense, cyclic knowledge graphs) and a dual-environment Reinforcement Learning (RL) framework (combining a Wikipedia-based simulator for stability with a real-world environment). The model reportedly achieves state-of-the-art results, outperforming all open-source agents and surpassing the 671B DeepSeek-V3.1 on "Humanity's Last Exam" (HLE).

The primary concerns (specifically from Reviewer uNmb) were that the paper resembled an "engineering report" rather than a scientific study. The authors successfully rebutted this by reframing the paper around specific research questions regarding data topology and environmental stability, leading the critical reviewer to raise their score from a 2 (Reject) to a 6. The current consensus is generally positive (scores: 8, 6, 6, 4). The authors are also committing to open-sourcing their dataset.

**Reviewer Concerns:**

Reviewer uNmb (Initial Score: 2): Argued the paper was an "engineering report" lacking a central scientific problem. Also confused the method with "Agentic CPT" (Continual Pre-training), which the authors clarified was not used. After the rebuttal, the Reviewer agree to change to 6.

Reviewer rnrd (Score: 6): This reviewer was positive but noted "missing pieces": lack of dataset statistics, lack of simulator validation, and unclear open-source plans.  The authors admitted they don't have a full study but argued that "training dynamics" correlate. So probably the reviewer would accept.

Reviewer Usyc (Score: 4): This reviewer was on the fence but had one major frustration: the paper claimed to go "beyond obfuscation" in its data generation but failed to define what that actually meant. The authors provided a very strong, specific answer to the "beyond obfuscation" critique.

Reviewer YWs5 (Score: 8): Strongly endorsed the paper, particularly the data construction and dual-environment approach.

**Reviewer Scores:**

Reviewer uNmb: 2->6. Reviewer rnrd 6->6/8. Reviewer Usyc: 4->6.

---

### Decision · Program_Chairs · 2026-01-26

Accept (Poster)